# Synaptic and circuit mechanisms prevent detrimentally precise correlation in the developing mammalian visual system

Ruben A Tikidji-Hamburyan[1]*, Gubbi Govindaiah[2], William Guido[2], Matthew T Colonnese[1]*

[1]Department of Pharmacology and Physiology, The George Washington University, Washington, United States; [2]Department of Anatomical Sciences and Neurobiology, University of Louisville, Louisville, United States

**Abstract** The developing visual thalamus and cortex extract positional information encoded in the correlated activity of retinal ganglion cells by synaptic plasticity, allowing for the refinement of connectivity. Here, we use a biophysical model of the visual thalamus during the initial visual circuit refinement period to explore the role of synaptic and circuit properties in the regulation of such neural correlations. We find that the NMDA receptor dominance, combined with weak recurrent excitation and inhibition characteristic of this age, prevents the emergence of spike-correlations between thalamocortical neurons on the millisecond timescale. Such precise correlations, which would emerge due to the broad, unrefined connections from the retina to the thalamus, reduce the spatial information contained by thalamic spikes, and therefore we term them 'parasitic' correlations. Our results suggest that developing synapses and circuits evolved mechanisms to compensate for such detrimental parasitic correlations arising from the unrefined and immature circuit.

**\*For correspondence:**
rath@gwu.edu (RAT-H);
colonnese@gwu.edu (MTC)

**Competing interest:** The authors declare that no competing interests exist.

## Editor's evaluation

The authors use detailed simulations to convincingly demonstrate that the temporal properties of synaptic transmission from retina to thalamus help to prevent short timescale correlations from hijacking the activity-dependent refinement of these circuits. These correlations are shown to be "parasitic" because although they can readily drive neural plasticity, they have little information about visual topography during the relevant period of refinement. This is an important point since it informs our understanding of activity-dependent development of neural circuits. The present study shows that it is not enough to simply posit that "neurons that wire together fire together," since some types of correlated firing are actually detrimental.

## Introduction

Correlated neural activity plays an essential role in circuit development and plasticity (*Katz and Shatz, 1996*). Gunther Stent, following Hebb's hypothesis, proposed that synchronized activity among presynaptic neurons allows them to effectively depolarize and fire the postsynaptic cell to maintain their synapses, while neurons that are not coordinated lose their connections (*Stent, 1973*). Mechanisms that generate synchronous spontaneous activity during initial circuit formation have been identified in every brain system (*Kirkby et al., 2013*). The auditory, touch, and visual circuits are organized as topographic maps in which sensory receptors project onto central nuclei in a manner that retains their relative spatial relationships, an organization maintained in thalamic and cortical regions. Such topography is initially established by chemotrophic cues which are reinforced and refined by

spontaneous activity in the sensory organ, which provides positional information because correlation among sensory inputs during development drops off with distance (*Katz and Shatz, 1996*). In the visual system, this activity takes the form of spontaneous waves in the retina that propagate in 2D space and 'synchronizes' the firing of nearby neurons, driving visuotopic refinement in the thalamus, superior colliculus, and cortex (*Seabrook et al., 2017*; *Huberman et al., 2008*).

While the mechanisms of correlated retinal activity generation are well studied, the central mechanisms by which this activity is processed and transformed into synaptic change and, ultimately, circuit structure are poorly understood. One crucial question is how the timescale of correlation influences developmental plasticity because timescale can determine the plasticity mechanisms potentially available to the developing neurons (*Zenke and Gerstner, 2017*; *Drew and Abbott, 2006*). In adults, thalamic and cortical neurons produce precise correlations on the timescale of milliseconds, optimal for visual processing (*Butts et al., 2007b*; *Usrey and Reid, 1999*). These correlations rely on precise refined connections and fast synaptic transmission (see Figure 3 for an insight) and, therefore, should be relevant only when the networks are refined and ready to operate in such timescales. Confirming this, experimental and theoretical work has shown that during development, topographic information in retinal waves is conveyed in coarse-grained correlation, maximal around 500ms and absent in time windows below 100ms (*Butts and Rokhsar, 2001*; *Butts et al., 2007a*), because the firing of ganglion cells within a wave is not precisely correlated (*Maccione et al., 2014*; *Stafford et al., 2009*). Early synapses make several adaptations to read-out such long correlation timescales and convert the information into synaptic changes and ultimately map refinement. These adaptations include synaptic currents with elongated decay kinetics to increase the integration window (*Hauser et al., 2014*; *Kleinschmidt et al., 1987*; *Fox et al., 1989*; *Hestrin, 1992*; *Taschenberger and von Gersdorff, 2000*), and burst-time-dependent plasticity with time-windows matched to timescales of topographic correlation in retinal waves (*Butts et al., 2007a*), a critical property for OF/OFF segregation (*Gjorgjieva et al., 2009*). However, if the correlation below the timescale of correlated activity in the retinal waves appears in descending pathways, it is likely to mask informative activity and damage developing networks as it has the potential to induce synaptic plasticity based on non-informative activation.

There are a number of reasons that such damaging precise correlations might arise within the developing brain. In the visual thalamus (dorsal lateral geniculate nucleus dLGN), each relay neuron receives functional inputs from 10 (*Jaubert-Miazza et al., 2005*; *Bickford et al., 2010*) to 20 (*Chen and Regehr, 2000*) nearby retinal ganglion cells (rGC) on postnatal days 7–10 (P7-P10), prior to refining to the 1–3 inputs seen in adults. Such polysynaptic convergence should cause correlation among relay neurons and their cortical targets with high temporal precision (*Sailamul et al., 2017*) causing the network to lose topographic information as outlined above. Calcium imaging indicates high correlations in the visual cortex during early development, which decorrelate rapidly around eye-opening (*Rochefort et al., 2009*; *Siegel et al., 2012*). However, timescale was not examined and these studies did not report inferred spike correlations. Polytrode recordings show high spike correlations at slow timescales in visual cortex and thalamus during retinal waves (*Colonnese et al., 2017*). However, rapid timescale correlations were very low suggesting some frequency-dependent filtering. Interestingly, fast-timescale correlations emerged after the period of retinal waves and eye-opening. The absence of spike correlations faster than the informative timescales for retinal topography – even in the face of the extensive convergence of the retinal axons and elevated correlations at informative timescales – raises multiple questions: Are there synaptic or circuit mechanisms that prevent relay neurons from synchronizing at timescales below the informative timescales of retinal waves? What functional advantage accrues by preventing the thalamus from precisely correlated activity?

Here, we address these questions using a biophysically detailed model of heterogeneous neurons in dLGN at P7-P10 driven by spike trains of retinal ganglion cells recorded *ex vivo* at these ages. Our computer simulations indicate that the promiscuous and imprecise synaptic connections observed during early development should drive rapid correlation among dLGN neurons, but these correlations are actively suppressed by the high ratio of NMDA-Receptor to AMPA-Receptor currents (NMDAR/AMPAR currents) present during early development. The emergence of a precise correlation is further prevented by the low levels of cortical and thalamic reticular nucleus input observed at these ages.

In total our results suggest a novel developmental principle: the properties of early synapses and circuits are tuned to eliminate potentially detrimental correlations that are the result of refining

networks because they could damage network formation. We term such correlations 'parasitic' as they actively leech positional information from the developing network. Suppression of such correlations is likely another factor guiding the evolution of developing synapses and circuits along with regulation of synaptic plasticity and optimizing integration time.

## Results

### Reproduction of the excitability and heterogeneity of thalamic relay neurons in a biophysical model

To model spike correlation in the developing thalamic network, we accurately reproduced neuron dynamics, synaptic properties, and network heterogeneity at the corresponding age. These models must be heterogeneous to avoid synchronization arising just because the neurons are identical; therefore, if spike correlation is observed, it would not be due to the homogeneity of neurons but rather the network properties. Even for relatively small networks, heterogeneity requires up to a few hundred different neuron models to populate the network. We used the standard evolutionary multiobjective optimization (EMO) approach to constrain model dynamics to match the dynamics observed in thalamocortical (TC) neurons recorded in a standard current-clamp protocol with step current stimulation *in vitro* at postnatal day 7–10 (P7-P10, see Experimental Procedures section for more details). The recordings were performed as routine TC neuron identification and characterization in previously published works: *Campbell et al., 2020* and *Govindaiah et al., 2020*. EMO methods yield up to a few hundred acceptable models for each recorded neuron (*Neymotin et al., 2017*; *Dura-Bernal et al., 2017*; *Iavarone et al., 2019*), allowing us to create a large, heterogeneous database of models that reproduce essential features of thalamocortical neurons at P7-P10.

We adopted a model of young adult (P14-18) somatosensory thalamocortical neurons in the ventrobasal thalamus developed by *Iavarone et al., 2019*. Even adult TC neurons are relatively electrically compact (*Sherman and Guillery, 2004*; *Bloomfield and Sherman, 1989*). Developing TC neurons have shorter and thicker processes (*Charalambakis et al., 2019*; *El-Danaf et al., 2015*), which allowed us to use a conductance-based 'pen-and-ball' two-compartment model with a single segment for the somatodendritic compartment and a multisegment compartment for an axon. All attempts to fit a model with a single somatodendritic compartment did not produce acceptable models, probably because axons and axon-hillocks are thicker and more electrically bound with the soma at this age. Therefore, the 'pen-and-ball' two-compartment model is the minimal model which sufficiently reproduces the behavior of the recorded neurons. We used both a well-established genetic algorithm with nondominated sorting (*Neymotin et al., 2017*; *Dura-Bernal et al., 2017*; *Deb et al., 2002*; *Deb, 2001*) and developed an in-house genetic algorithm with Krayzman's adaptive multiobjective optimization EMO(s) to fit dynamics of somatic voltage recorded in the current-clamp protocol. Both EMO methods yielded similar quality and quantity of acceptable models. However, a single optimization method could potentially bias models to specific parameter- or feature-regions; therefore, we selected between each EMO method randomly for each run to avoid such bias. Details of the second method are given in the Appendix Genetic algorithm with Krayzman's adaptive multiobjective optimization (KAMOGA).

Because neuron geometry, expression of specific subunits for ion channels, and densities of these channels change during development, in addition to the standard free parameter set for fitting, such as channel densities (conductance), we allowed minor adjustment in additional classes of model parameters in the somatodendritic compartment, such as:

- intracellular calcium buffer depth,
- calcium pump rates,
- reversal potentials for sodium, potassium, and nonselective voltage-gated cation channels (h-channel, HCN),
- the time constant for the calcium-activated potassium channel (SK),
- half-points of Boltzmann's steady-state functions,
- the geometry of both compartments – the length of the somatic compartment and the length and diameter of the axonal compartment.

Thus, in total, the neuron model had 29 free parameters for optimization, listed in the methods section Neuron Optimization Pipeline and supplementary dataset for *Figure 1*. The list of objective

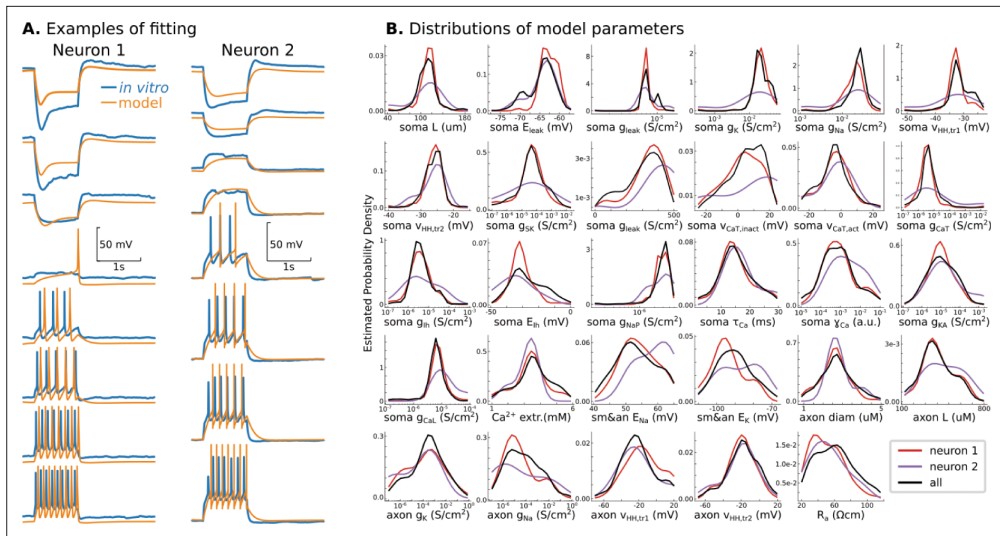

**Figure 1.** Conductance-based model of P7 TC neurons. (**A**) Randomly selected examples for two recorded neurons in the database. (**B**) Probability density distributions of all 29 model parameters, obtained as Gaussian estimator over all models in the database. Note that EMOs use a logarithmic scale for some parameters, and therefore, probability density distributions are also estimated in the logarithmic scale. Black lines are PDFs for all models in the database, and color lines indicate PDFs for models fitted to recorded neurons shown in **A**.

The online version of this article includes the following source data and figure supplement(s) for figure 1:

**Figure supplement 1.** Principal component analysis shows the separation of models fitted to the same recorded neurons as in **Figure 1**.

**Figure supplement 1—source data 1.** Data base with 286 models of TC neurons at P7 (available on Zenodo).

functions for fitting and the description of complete pipeline fitting-validation-evaluation is also given in the same Materials and methods section.

Both EMOs yield from 20 to 100 acceptable but different models for each recorded neuron, similar to the outcomes reported by *Iavarone et al., 2019* and *Neymotin et al., 2017*. Therefore, these fitting procedures allow us to accumulate a few thousand models in our current database. *Figure 1A* shows two models for two different recorded neurons, randomly chosen from the database. A reduced, fully validated, and human-evaluated database of 286 models, enough to reproduce any results in this paper, can be found in the supplementary data for *Figure 1*.

Somatic voltage dynamics in the current-clamp protocol may be insufficient to specify the unique contributions of different ion channels to the dynamics of a conductance-based model. As shown previously by Marder and colleagues, a single-compartment conductance-based model can have similar dynamics for two sets of parameters with extreme differences in the contribution of different channels (*Marder, 2011*; *Prinz et al., 2004*; *Prinz et al., 2003*). Because similar model dynamics can result from different distinct configurations of channel conductance(s), it is possible that EMO, extensively searching for model configurations, might produce acceptable models in distinctly different regions of the parameter space. If this occurred we expect to find multimodal distributions of model parameters with peaks at regions where model dynamics match the target dynamics of the recorded neuron. In this situation, choosing which cluster of models best represents real neurons is difficult and would undermine confidence in our neuron models. To examine the distributions of our generated models, for each recorded neuron, we ran from 2 to 20 independent EMOs, each starting from the random set of points in the parameter space. Although the dynamics of TC neurons are different, parameters in the database do not show multimodal distributions (*Figure 1B*). To our surprise, there is a single region for all our TC cells, and therefore distributions of individual parameters are mono-modal. Principal component analysis of parameters in the resulting database shows that all models occupy a relatively compact single region of the PCA space, suggesting a unique parameter region that matches TC neurons at P7-P10. Moreover, with a few exceptions, the models fitted to the different recorded neurons separate well in the principal component space (*Figure 1—figure supplement 1*). Surprisingly, PCA indicates distinctive correlations of model parameters (PCA features) that

characterize each recorded neuron, suggesting that EMOs achieve precision when recorded neurons can be distinguished by the parameters of their models, that is, differences between recorded neuron dynamics lead to separation of model parameters in the PCA space.

Thus, unless a number of the parameters *in vivo* lie outside the parameter ranges considered, we conclude that our database is a consistent and likely valid representation of the real dLGN TC neurons at the specific point of their development. We expect the models have somewhat greater heterogeneity than real neurons due to the inability to determine a single exact parameter set for each recorded neuron.

## NMDA receptor dominance at developing retinothalamic synapses decorrelates relay neurons in millisecond timescale

We first examined the role of glutamatergic synaptic currents in the regulation of the correlation of thalamocortical neurons during the P7-10 period. As at many developing synapses, NMDA receptors provide the dominant conductance, even comprising the only receptor type in new synapses (*Rumpel et al., 1998*; *Chen and Regehr, 2000*; *Shah and Crair, 2008*). NMDA receptors are an established regulator of synchronization in adult networks (*Jacobsen et al., 2001*) and, thus, likely to play a critical role at these ages.

To systematically study the role of synaptic currents in spike correlation, we used our database to construct a small model of the dLGN network with realistic heterogeneity and drove the network by rGC cell spikes recorded *ex vivo*, obtained from the *waverepo* repository of early retinal activity (*Eglen et al., 2014a*; *Eglen et al., 2014b*). To provide sufficient spatial sampling and resolution, we selected recordings of P6-P10 wild-type mice, recorded with a high spatial-resolution electrode array and at least 30 adjacent active electrodes *Maccione et al., 2014* and *Stafford et al., 2009*. Recurrent inputs from the visual cortex provide an indispensable excitation to dLGN at this age (*Murata and Colonnese, 2016*), and inhibitory inputs from the thalamic reticular nucleus (TRN) are also likely to be active (*Evrard and Ropert, 2009*; *Minlebaev et al., 2011*). These inputs were modeled as non-specific feedback in the sections below and ignored here to examine the NMDA role directly. While present in the dLGN at this age, intrinsic interneurons are not reliably driven by retinal input before eye-opening (*Bickford et al., 2010*), and so we did not include them in the model.

To set the synaptic parameters for the retinogeniculate synapse, we used the ratio of peak-to-peak AMPA to NMDA currents $I_{AMPA}/I_{NMDA} = 0.78 \pm 0.09$ according to the closest published measurement (P7) (*Shah and Crair, 2008*). This observation allows estimation of the ratio of peak-to-peak conductance for these currents at $g_{NMDA}/g_{AMPA} \approx 2.25$ (see details in the dLGN Network Model method section). The retinogeniculate synapses also exhibit a paired-pulse ratio of 0.73 (*Chen and Regehr, 2000*), which we model using the simplified Tsodyks–Markram model (*Tsodyks et al., 2000*). The single RGC fiber synaptic conductance is unknown, though it appears to be less than required to fire a TC neuron (*Dilger et al., 2011*; *Liu and Chen, 2008*). Because our model of the dLGN network consists of a heterogeneous population of neurons, randomly chosen from the database, there is no single value for synaptic conductance which could satisfy all neurons in the model. The same value for synaptic conductance can be subthreshold for some neurons in our database but drive others into a depolarization block. Therefore, it is preferable to use a functional criterion to set synaptic conductance. We used the mean firing rate recorded *in vivo* at this age as our functional criterion. The mean firing rate of TC neurons is around 1 spike/s (*Murata and Colonnese, 2018*), but because silencing the cortex and abolishing the corticothalamic feedback reduces this rate by 50% (*Murata and Colonnese, 2016*) in models without cortical feedback, target firing rate should be set to 0.5 spike/s. Instead of applying EMO methods to fit the synaptic conductance for each individual neuron in a heterogeneous dLGN network with randomly set connection from rGC neurons, we assumed that total synaptic conductance is under the control of a homeostatic process with a target set-point of average firing rate (0.5 spikes/s for this model) and allowed homeostasis to regulate the firing rate at the level of individual neurons. Although an extensive body of evidence supports homeostatic regulation, which affects TC neurons' excitability and the number and conductance of input synapses in the developing dLGN (*Tien and Kerschensteiner, 2018*), the homeostatic process in our model regulates only total synaptic conductance. Thus, the introduction of the homeostatic process should be considered as a biologically inspired algorithm that allows to meet the functional criteria of mean firing rate with preserved heterogeneity of the

network, rather than a model of much more complex homeostatic processes ongoing in the developing thalamus.

During the first 3–4 weeks of development, the functional convergence of rGC axons on single relay neurons is reduced from >20 to 1–3 (*Guido, 2018*; *Liang and Chen, 2020*). The estimates of how many rGC axons converge to a single TC cell at P7-P10 vary from 10 (*Jaubert-Miazza et al., 2005*; *Bickford et al., 2010*) to 20 (*Chen and Regehr, 2000*). Because convergent inputs are most likely to come from adjacent rGC (*Liang and Chen, 2020*), we organize retinogeniculate synapses such that both the connection probability and synaptic conductance have Gaussian dependence on the distance between rGC and TC neurons, with the degree of convergence set by the parameter $\sigma$. We test a range of $\sigma$ from 1, which corresponds to adult convergence (1–3 rGC connections per TC neuron), to 9 (20 rGC per TC neuron; $\sigma = 4$ generates 10). For each $\sigma$, ten network models are generated for analysis. Each simulation was run until the total glutamatergic conductance reached the steady-state region, and the mean population firing rate reached the homeostatic set-point of 0.5 spike/s with 10% accuracy. We computed the spike correlation in each model from the last 20 min of network activity (when synaptic weights were stabilized). Correlations were computed as in previous experimental studies (*Colonnese et al., 2017*): the spike train of each neuron is convolved with a Mexican-hat-like kernel, and the Pearson correlation coefficient is computed for each pair of neurons within the population as described in Quantification of spike correlation section. For our initial examination of the effects of rGC convergence and synaptic currents we selected a short-timescale by using a kernel which is positive in a 48ms window (see the same methods section), below that informative for topography and shown to be functionally minimal at these ages (*Colonnese et al., 2017*).

When NMDAR/AMPAR ratios are set to biologically accurate levels, the distribution (*Figure 2A2*) and mean (*Figure 2A3*) of spike correlations are very low and in good agreement with experimental observations for these ages (*Colonnese et al., 2017*). Mean correlations are around 0.02–0.03 for expected levels of convergence $\sigma = [4, 9]$ at these ages. The distribution of correlation was centered near zero, and some pairs displayed positive correlation, as observed in the thalamus. While biologically accurate, the level of correlation is unexpectedly low given the levels of input convergence and showed a surprising insensitivity to the convergence parameter $\sigma$. With increasing convergence, neurons receive more synaptic inputs from the same rGCs. Therefore, the spike sources of the synaptic currents overlap, and the overall activation of the population becomes more homogeneous. For the range of convergences $\sigma = [1, 9]$, the number of inputs varies from 1-3 to 20 and should dramatically increase correlation within the population. Even more surprising, such a low correlation does not result from population heterogeneity. Homogeneous networks populated with the same randomly chosen TC neuron model from the database show a similar low correlation and insensitivity to the convergence (*Figure 2—figure supplement 1*, left) suggesting that another factor is active in reducing correlation at this timescale.

We hypothesized that the low correlation and convergence-insensitivity might result from the NMDA receptor dominance at immature synapses. To test this, we examined a network with only fast AMPA receptors at the rGC synapse. In this case, we observe an approximately 20-fold increase in mean spike correlations, which reached 0.3–0.4 (*Figure 2B*) at biologically relevant levels of $\sigma$. With the fast AMPA receptors, the convergence factor $\sigma$ controls spike correlation in these networks (*Figure 2B2 and B3*) as expected. As a result of this control, the mean spike correlation is relatively low for the adult convergence ($\sigma = 1$) and shows a narrow distribution with a peak slightly shifted from zero, indicating that fast AMPAR-dominant current cannot induce significant fast-timescale correlation beyond what present in the input and therefore fast synaptic transmission is safe-to-operate in adult thalamus. Moreover, models with only fast AMPA receptors are sensitive to model heterogeneity (*Figure 2—figure supplement 1*, right). Although correlation in homogeneous networks is approximately the same for adult convergence, variability quickly increases as convergence approaches the number of connections in early development and can be more than twofold for $\sigma = 9$; and therefore, as expected, heterogeneity is the required and critical property of our models which allows avoiding under- or overestimation of correlations in the network. Overall, we find that glutamate synapses with dominant AMPA receptors allow the high-levels of convergence present during the early stages of development ($\sigma = [4, 9]$) to induce fast correlations but do not themselves impose additional correlation (for example in the case of the adult convergence). We conclude the level of fast correlation in the heterogeneous network of P7-P10 TC neurons depends on the composition of synaptic currents and

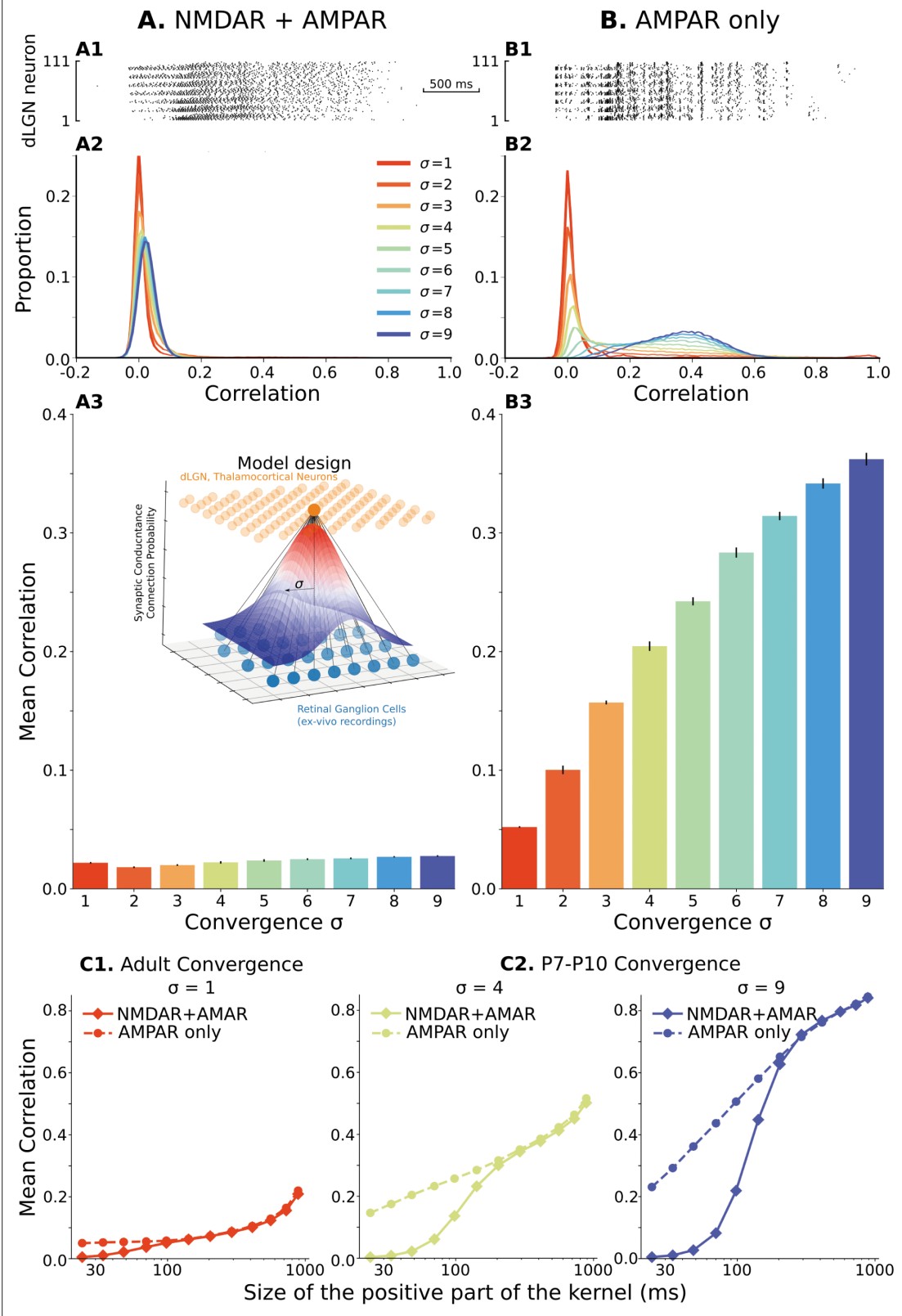

**Figure 2.** The model of the dLGN network at P7-P10 activated by spikes of rGC recorded *ex vivo*. (**A**) Retinogeniculate synapses have both NMDA and AMPA currents, with NMDA dominance. (**B**) The same models with NMDA currents disabled. **A1** and **B1** are examples of two models ($\sigma = 4$) responding to the same retinal wave. **A2** and **B2** are mean distributions of pairwise spike correlation for different convergence factors ($\sigma$). **A3** and **B3** mean and standard deviation of network-wise average pairwise spike correlation for ten network models for each type of the synaptic drive and

*Figure 2 continued on next page*

*Figure 2 continued*

convergence factor ($\sigma$). (**C**) A mean network correlation depends on the timescales of the correlation. Dependence on the timescale is shown for adult convergence ($\sigma = 1$, **C1**) and two estimates of P7-P10 convergence (**C2**): 10 inputs per TC neuron ($\sigma = 4$, left) and 20 inputs per TC neuron ($\sigma = 9$, right). The horizontal axis is the maximal $\Delta t$ between two spikes, which results in a positive correlation for a given kernel size. The insert schematically shows the model design. For all C plots, mean is computed over ten independently generated and initialized network models.

The online version of this article includes the following figure supplement(s) for figure 2:

**Figure supplement 1.** The same as in *Figure 2A3 and B3* but for homogeneous populations.

**Figure supplement 2.** Network correlation dependes on proportion of NMDAR and AMPAR currents and convergance.

**Figure supplement 3.** Two families of curves which show the same dependeces as *Figure 2A3 and B3* for 12 different kirnels shown in legend.

**Figure supplement 4.** Decorrelation results from interactions between intrinsic neuron dynamics and slow NMDAR current.

convergence of retinal inputs (*Figure 2—figure supplement 2*). These results show that slow NMDA receptor currents can decorrelate neural firing in a pure feedforward network, a phenomenon that is well established for recurrent networks by *Pinsky and Rinzel, 1994*, and may explain why thalamic and cortical activity is so poorly correlated *in vivo* at rapid timescales at these ages (*Colonnese et al., 2017*), despite massive co-activation by the poorly refined retinal inputs (*Jaubert-Miazza et al., 2005*; *Bickford et al., 2010*; *Chen and Regehr, 2000*).

If the NMDA receptor current is tuning the network to timescales appropriate for development, dLGN networks with age-appropriate NMDAR dominance should not desynchronize spike correlation at the timescales which convey spatial information present in retinal waves (i.e. >100ms, *Butts and Rokhsar, 2001*) To test this, we examined the effects of different timescales at critical levels of convergence. We scaled the positive and negative components of the Mexican-hat kernel proportionally such that the kernel is positive in a window ranging between 20ms and 1 s (*Figure 2C* and *Figure 2—figure supplement 3*). For both adult and immature levels of convergence, a high NMDAR/AMPAR ratio causes decorrelation of the network specifically at fast-timescales (*Figure 2C*) while AMPAR only synapses scale approximately logarithmically with time window width. Note that with a constant number of rGC and TC neurons, the divergence of retinal connections is proportional to convergence of rGC connections on TC neurons. For adult convergence (σ=1, *Figure 2C1*), NMDAR dominance reduced correlations when windows were below 100ms, while for immature levels of convergence (σ=4 and 9) NMDAR-dominant network diverged at timescales below 150–200ms in line with the observed timescales of topographic information. Thus, high levels of immature NMDA receptors appear to specifically tune the thalamic network to correlations at informative timescales by suppressing correlations likely to contain little information.

We briefly examined the mechanism of NMDAR-dependent desynchronization by determining if it is entirely dependent on the low-pass filtering caused by the slow kinetics of immature NMDARs (*Hauser et al., 2014*; *Constantine-Paton et al., 1990*), or on the interaction of NMDAR characteristics with neuron dynamics. To test this, we sped up and slowed down the neuron intrinsic dynamics by manipulating the temperature of the model, while the dynamics of NMDAR currents were held the same as before. Sufficient adjustment of neuron dynamics increases fast correlations in both cases (*Figure 2—figure supplement 4* top), showing that the interaction between neuron dynamics and NMDARs is driving the desynchronization. Adjusting NMDAR dynamics to adult decay times ($\tau_{decay} = 74$ms, *Figure 2—figure supplement 4* bottom left) similarly increased fast correlations. In total, our results suggest that decorrelation results from an interaction of slow intrinsic dynamics of immature neurons and NMDAR current decay times.

## Precisely correlated dLGN spikes lose retinotopic information

Precise correlations exist in the adult dLGN, where they are driven by the relatively large divergence but small (1-3) convergence of rGCs on relay cells as well as by visual features (*Butts et al., 2007a*; *Alonso et al., 1996*). Such correlations are however relatively rare and limited to very close relay cells. When convergence is high and driven by fast, adult-like synapses we observed high-levels of precise correlation among most neurons in a given region very different to sparse synchronization in adults. Because of the dynamics of retinal waves, such fast correlations cannot inform synaptic refinement for topography or ON/OFF response segregation. It is likely that strong correlations that are non-informative would disrupt circuit formation as they would strengthen synapses randomly, not

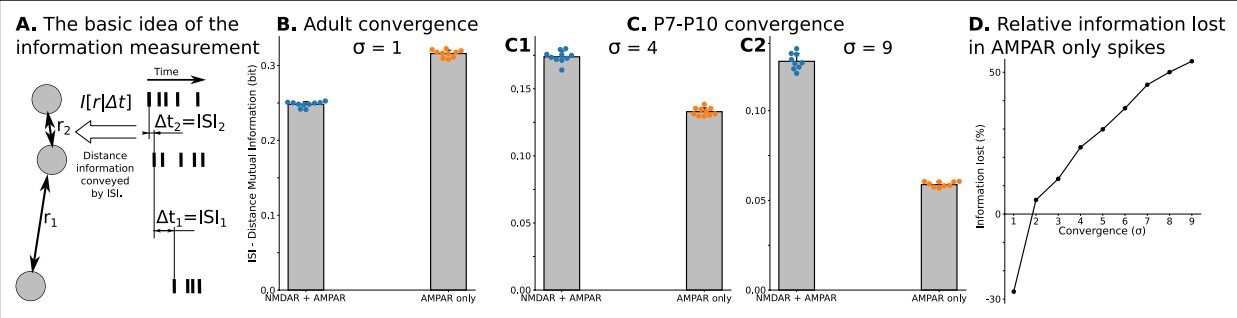

**Figure 3.** Spatial information encoded in interspike/interburst intervals (ISI) of TC neurons in the dLGN model. For B and all C plots, mean and standard deviation is computed over ten independently generated and initialized network models. (**A**) Mutual information $I[r, \Delta t]$ as a quantitative measure for predictability of distances $r$ from observed ISI $\Delta t$. (**B**) Mutual information $I[r, \Delta t]$ in a model with adult-like convergence ($\sigma = 1$). (**C**) The same as in B but for two estimates of P7-P10 convergence: 10 inputs per TC neuron ($\sigma = 4$, **C1**) and 20 inputs per TC neuron ($\sigma = 9$, **C2**). (**D**) Dependence of information lost/gained in spikes of models with AMPA receptors only compared to models with NMDA + AMPA mixture of receptors on the convergence of rGC inputs to a single TC neuron ($\sigma$). For B, and all C plots, mean and standard deviation is computed over ten independently generated and initialized network models.

specifically as required for circuit refinement. However, such correlations could be detrimental in an additional way, reducing the topographic and ON/OFF information present at slower timescales, in effect acting parasitically to leech information from the developing network.

To examine the effect of precisely correlated spikes on spatial information, and determine if they would be expected to aid or disrupt the refinement of thalamocortical connections, we applied classical information theory analysis suggested by *Butts and Rokhsar, 2001* to spikes of TC neurons in our model of dLGN network. The essence of this approach is to measure mutual information between interspike/interburst intervals (ISI, $\Delta t$) for spikes of two neurons and distance ($r$) between these two neurons ($I[r, \Delta t]$), which quantifies information about distances conveyed by neuron spiking (*Figure 3A*, see Materials and methods section Quantification of mutual information). We computed mutual information $I[r, \Delta t]$ for dLGN models with different convergence factors ($\sigma$) described in detail in the previous section. *Figure 3B and C* shows the effect of synaptic current composition on spatial information in the spikes of TC neurons. For adult convergence ($\sigma = 1$), spikes of neurons driven by pure AMPAR currents convey more spatial information than those driven by the mixture of NMDA and AMPA receptors found in developing dLGN. In contrast, for both estimates of P7-P10 convergence: 10 inputs per TC neuron (*Figure 3C1*) and 20 inputs per TC neuron (*Figure 3C2*) spatial information is much higher for the biological mixture of synaptic currents than with AMPARs only.

We qualify information loss or gain with AMPAR-only synapses as follows:

$$I_{\text{lost}} = \frac{<I[r, \Delta t]_{\text{NMDAR+AMPAR}}> - <I[r, \Delta t]_{\text{AMPAR only}}>}{<I[r, \Delta t]_{\text{NMDAR+AMPAR}}>}$$

where $< >$ denotes an average value over 10 trials. This analysis (*Figure 3D*) shows that AMPAR-only synapses are advantageous only for adult convergence. For P7-P10 convergence, models with AMPAR-only synaptic currents lose from 25% to more than 50% of spatial information present with the natural composition of NMDAR and AMPAR currents. Overall these results show that precise correlations induced by high-convergence (if not suppressed by NMDARs) during refinement are not only non-informative, but actually cause thalamic relay neurons to lose information conveyed to cortex, in this way acting in a parasitic fashion that must be suppressed to avoid delay or degradation of the refinement process.

## Role of the thalamic reticular nucleus in precise spike correlation in dLGN

So far, our modeling has only considered the role of the feed-forward, driver inputs to dLGN. In adults, however, synchronization of the thalamocortical circuit is strongly influenced by the feedback excitation and inhibition provided by corticothalamic neurons (CT) and thalamic reticular (TRN) nucleus (*Crunelli et al., 2018*; *McCormick et al., 2020*; *Pinault, 2004*). While the development of these projections are delayed relative to the retinal and thalamocortical projections, they are functionally

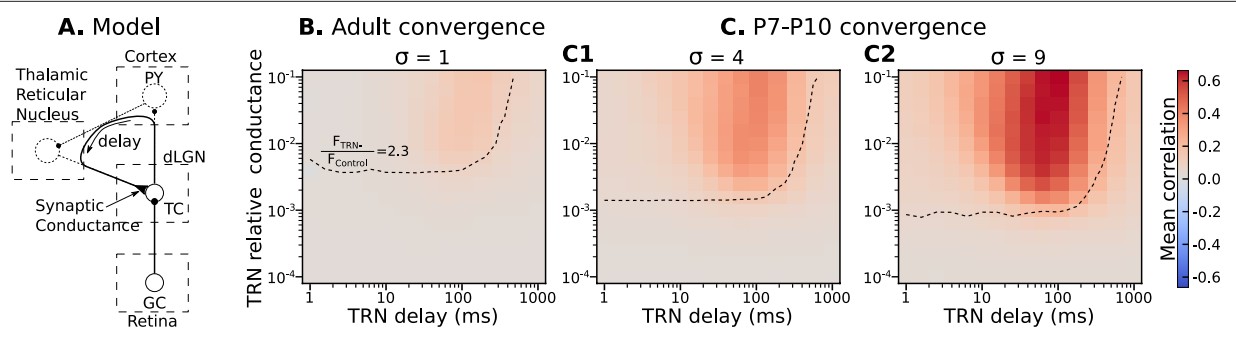

**Figure 4.** Parameters of TRN inhibitory feedback, that match *in vivo* observations, are outside the range when they can induce precise correlation in TC neurons. (**A**) Schematic of the model design. (**B**) Effect of different levels of TRN relative inhibitory conductance (ordinate) and delay (abscissa) on mean spike correlations in a model with adult-like convergence of rGC inputs to a single TC neuron ($\sigma = 1$). The estimated effect of TRN silencing on firing rate is shown as a dashed line. (**C**) The same as in B but for two estimates of P7-P10 convergence: 10 inputs per TC neuron ($\sigma = 4$, **C1**) and 20 inputs per TC neuron ($\sigma = 9$, **C2**).

connected at the ages examined and so may influence correlation and synchronization during waves. We first examined the effects of recurrent inhibition as provided by the TRN. Unfortunately, to our knowledge, direct measurements of TRN→dLGN synaptic conductance and delays do not exist for these ages. However, TRN makes recurrent connections with VPM at the ages of interest (*Evrard and Ropert, 2009*), and inhibiting the firing of parvalbumin expressing neurons in the TRN at P9-P10 increases dLGN neuron firing by 2.3 fold (MTC, unpublished data). To accommodate ambiguity in the circuitry leading to recurrent inhibition we systematically studied parameter spaces for synaptic conductance and delay to gauge the full potential of this projection to influence dLGN activity.

To examine the potential effect of TRN inputs on precise spike correlation, we modeled them as non-specific (to all neurons) inhibitory synapses. Again, ten different models with the same biological NMDA/AMPA conductance ratio of 2.25 were created for each conductance-delay pair. Each model ran until the mean firing rate of the population reached the homeostatic set-point 0.5 Hz ±10%. We study an exhaustive range of conductance-delay pairs (delays [1ms, 1 s] and relative synaptic conductance [$10^{-4}$, 1]), many of which are likely far from biologically plausible values. To determine which pairs result in firing changes to dLGN neurons similar to that observed *in vivo* when TRN is silenced, we disabled both the homeostasis and TRN feedback in models that reached homeostatic steady-state and measured firing rates to find the contour line where this ratio matches the experimental value $F_{TRN-}/F_{Control} = 2.3$. These contour lines overlapped on correlation heatmaps are shown in *Figure 4B and C*.

For adult convergence $\sigma = 1$, inhibitory TRN connections do not drive significant correlation under any sampled conditions. Even for strong and significantly delayed inhibitory currents, the mean spike correlation remained below 0.1 (*Figure 4B*). Therefore, this feedback is 'safe' and cannot corrupt information conveyed in the fine-grain timescales in adults (*Butts et al., 2007b*).

By contrast, at P7-P10 convergence levels ($\sigma = 4$ or $\sigma = 9$), some combinations of high-conductance/long-delay (>10ms) increase mean spike correlations to levels similar to the AMPAR-only condition (0.6 and higher) (*Figure 4C1 and C2*). However, none of the parameter pairs that resulted in considerable correlations matches an increase in the firing rate $F_{TRN-}/F_{Control} = 2.3$ observed *in vivo*. This modeling suggests that the delayed development of functional TRN synapses, as suggested by *Murata and Colonnese, 2016*, is desirable partly because strong TRN synapses early in development (particularly slow ones) could re-introduce parasitic correlations that result from the immature high convergence.

### Role of the combined cortical and thalamic reticular nucleus feedback on precise spike correlation in developing dLGN

Although realistic levels of TRN inhibitory synaptic conductance are not sufficient to induce high spike correlations in the dLGN networks, there is still the possibility that the excitatory cortical feedback (possibly in combination with inhibitory inputs from TRN) can increase spike correlation of TC neurons. To test whether a combination of excitation and inhibition can correlate TC neuron spiking, we add an excitatory feedback loop that represents the cortical effect on dLGN (*Figure 5A*).

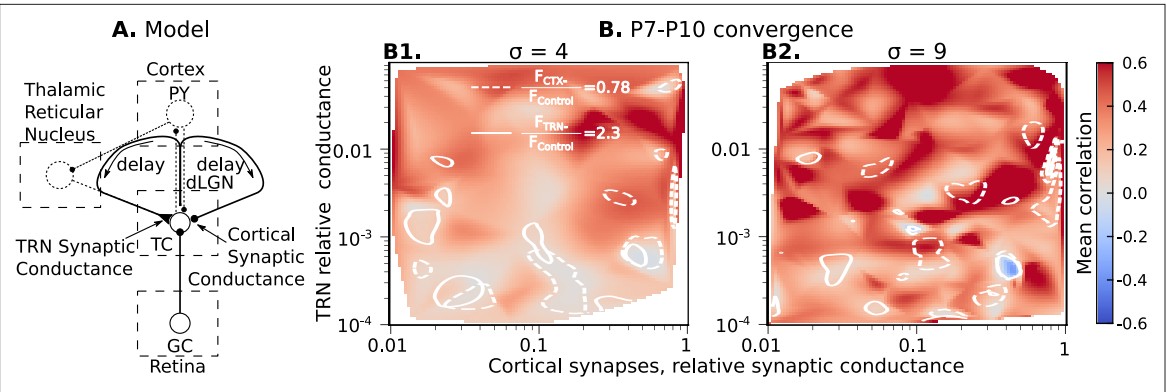

**Figure 5.** Cortical excitation and TRN inhibition parameters that match *in vivo* observations are outside values where they can synchronize TC neurons. (**A**) Schematic of the extended model. (**B**) Dependence of mean network correlation upon TRN and cortical relative synaptic conductance. Two heatmaps with the convergence of 10 inputs per TC neuron ($\sigma = 4$, **B1**) and 20 inputs per TC neuron ($\sigma = 9$, **B2**) are shown. White solid lines and white dash lines indicate mean changes in firing rate, which match the observed TRN silencing ($F_{TRN-}/F_{Control} = 2.3$) and cortex silencing ($F_{CTX-}/F_{Control} = 0.78$) *in vivo* experiments, correspondingly. These show the regions of possible conductance based on *in vivo* values.

The online version of this article includes the following figure supplement(s) for figure 5:

**Figure supplement 1.** Heatmaps for mean spike correlation (left set), $F_{TRN-}/F_{Control}$ (middle set), and $F_{CTX-}/F_{Control}$ (right set) for all four model parameters: TRN synaptic conductance $g_{TRN}$, TRN delay, CTX synaptic conductance $g_{CTX}$, and CTX delay.

**Figure supplement 2.** The same as in *Figure 5—figure supplement 1*, but for $\sigma = 9$.

As for TRN, exact measurements of CT feedback conductance and delay do not exist. Therefore, we evaluate the roles of cortical synaptic conductance ($g_{CTX}$), cortical feedback delay, TRN synaptic conductance ($g_{TRN}$), and TRN feedback delay to dLGN synchronization by treating each of them as free model parameters. For each parameter set, ten models were generated as above, and simulation for each network model runs until the mean firing rate of the population reaches the homeostatic set-point. In this case, the set-point was 1 spike/s, the level observed *in vivo* when CT feedback is intact. In these simulations, homeostatic regulation is heterosynaptic (*Fiete et al., 2010*; *Wu et al., 2020*) and regulates both rGC and CT synaptic strengths simultaneously. However, it does not change a ratio between strengths of rGC and CT inputs; therefore we report the relative CT conductance in *Figure 5B*.

With four independent axes, the model parameter space is too big to explore systematically on available computational resources. Instead, we used Monte-Carlo sampling and reconstructed maps of mean spike correlation, mean change in firing rate when TRN is silenced $F_{TRN-}/F_{Control}$, and mean change in firing rate when the cortex is silenced $F_{CTX-}/F_{Control}$ (*Figure 5—figure supplements 1 and 2*). Although the parameter space is four-dimensional, we found two critical parameters to primarily define necessary conditions for TC spike correlation: conductance of the TRN and CT synapses. We project all sampling points onto the two-dimensional conductance map to simplify the visual representation. To visualize regions where the models generated CT and TRN inputs that, when blocked, have similar amplitude effects on firing, we overlap contour lines for $F_{TRN-}/F_{Control} = 2.3$ and $F_{CTX-}/F_{Control} = 0.78$ on the heatmaps of mean spike correlation. This analysis shows that while there are parameters of CT and TRN inputs that generate high correlation even in the presence of NMDA receptor dominance, these invariably lay outside the intersections of the two contour lines. Therefore, combinations of synaptic conductance that reproduce the *in vivo* data are insufficient for network synchronization. These results support our hypothesis that the developing thalamocortical system avoids configurations that would drive precise correlation resulting from the exuberant retinal inputs.

## Discussion

In this study, we modeled some of the key synaptic and circuit factors that regulate the precision of spike correlations in the developing dLGN. By employing both well-established and novel methods of multiobjective evolutionary optimization, we reconstructed the dynamics of thalamocortical neurons in dLGN at postnatal days 7–10. We constructed a heterogeneous network composed of these

realistic cellular models, which represents the dLGN network at a specific developmental time point during initial circuit formation. We drove this network with spikes of retinal ganglion cells recorded *ex vivo* at similar ages to identify a previously unexpected role for NMDARs during development. Beyond their expected role in synaptic plasticity and amplification of weak synapses, we showed that NMDARs actively prevent the correlation of thalamic relay neurons on the millisecond scale. Such precise synchronizations are generated by the extensive convergence of the refining retinal ganglion cells at this age, however they do not provide relevant information to the developing system, which is conveyed only at longer timescales (100ms and above, *Butts and Rokhsar, 2001*). Using an information-theoretic approach, we show that these precise correlations are not simply non-informative, but that they actually reduce topographic information conveyed by thalamic neurons to the cortex. We propose that such correlations, arising as a consequence of the poorly refined immature network, can be considered 'parasitic', analogous to 'parasitic capacitance' (or other parasitic circuit elements) in electronics that drain 'vitals' important to circuit function and 'parasitic vortices' in aviation which reduce elevation force and create an additional drug, unless eliminated by curving the end of the airplane wings. We further showed that if adult-like connectivity from the feedback projections of cortex and TRN were present, they would reinstate parasitic correlations. Instead feedback connection strength appears to be maintained at a low level until more adult-like levels of convergence are attained, minimizing the emergence of the parasitic correlation.

These results allowed us to hypothesize that the desynchronization of thalamic neurons by NMDARs is an important component of early activity and that the developmental delays in cortical and TRN connectivity are likely selected by evolution to avoid the generation of precise, rapid-timescale, parasitic spike correlations in TC neurons. Overall, our results suggested a novel developmental principle that just as appropriate synchronization must be generated in inputs and their targets to refine synaptic connectivity (*Thompson et al., 2017*; *Feller, 2009*), parasitic synchronization arising 'accidentally' from the exuberant connectivity present during early development must be minimized to maximize information transfer and prevent premature or aberrant plasticity in the downstream circuits.

## Why can precise correlation be detrimental? Dual effects of correlation during development

In the developing visual system, spontaneous waves of activation correlate activity of nearby ganglion cells (*Wong et al., 1993*). The spatial information supplied by this correlation is used to reduce exuberant connectivity and refine topographic mapping in the primary retinal targets as well as the visual cortex (*Huberman et al., 2008*). Genetic manipulations that increase the size and intensity of retinal correlation reduce the refinement of retinal projection in the thalamus and superior colliculus (*Seabrook et al., 2017*). Because waves propagate slowly and only poorly correlate the firing of rGCs, topographic information is conveyed by the developing retina only at timescales greater than 100ms (*Butts et al., 1999*; *Butts et al., 2007a*; *Butts and Kanold, 2010*). The reason for this slow propagation is that the wavefront is generated by the volume release of acetylcholine and its bulk diffusion (*Ford et al., 2012*). Why nearby rGCs are poorly correlated during waves is poorly understood. While they maintain a high firing rate (~10 Hz) during a wave, this firing appears random and not correlated on a fast timescale to nearby rGCs, likely because intraretinal circuits mediating ganglion cell interaction (lateral inhibition, etc) are not yet formed (*Tian, 2004*). Within the area of the cortex driven by a single wave, neurons maintain a diversity of neural receptive fields, including multiple orientations, direction selectivity, and even visual responsiveness, which is critical for visual processing. These diverse receptive fields emerge from refined thalamic as well as recurrent local cortical inputs (*Niell and Scanziani, 2021*). By eliminating correlations on timescales below those that convey relevant information at a particular age and by preserving the correlations in coarse time scales, the synaptic and circuit mechanisms we identify, allow for topographic refinement and stabilization before the emergence of a diversity of receptive fields within an area of visual space. We did not explicitly model retinal dynamics present during mature vision, however our results with low convergence are consistent with the notion that with decreasing NMDAR dominance and decay times (*Liu and Chen, 2008*), as well as the formation of recurrent thalamic connections (*Guido, 2018*), the ability to support fast synchronization increases as visual information that uses such rapid timescales develops after eye-opening (*Usrey and Reid, 1999*; *Butts et al., 2007b*). Our results suggest a novel mechanism by which critical periods can be regulated: the changing timescales of synaptic currents and feedback connections

modifying the time and degree of correlation. For example, dominant NMDAR currents should be detrimental after vision onset because the informative correlations below 10ms are important in adult dLGN spikes (*Butts et al., 2007b*). This may play a role in the later corticothalamic critical period, when reduced corticothalamic input decreases retinothalamic refinement (*Hooks and Chen, 2020*), which our results suggest would reduce fast, local synchronization important for precise refinement.

## Multiple roles for NMDA receptors during development

An essential role for NMDARs in the segregation and refinement of glutamatergic afferents has been demonstrated in multiple species and sensory systems (*Ewald, 2009*). This role has largely been ascribed to their capacity to induce synaptic plasticity in response to synchronous activity. However, NMDARs are not critical for synapse stabilization, sprouting, or many forms of afferent refinement and segregation (*Iwasato et al., 2000*; *Hahm et al., 1991*; *Colonnese and Constantine-Paton, 2001*; *Huang and Pallas, 2001*), likely because other forms of calcium entry can drive the necessary synaptic plasticity for afferent refinement (*Lee et al., 2014*; *Kuo and Dringenberg, 2012*). In addition to the regulation of synaptic plasticity, NMDA receptors play important roles in network activity. NMDA currents amplify activity at the retinogeniculate synapse both through their long decay times (*Liu and Chen, 2008*) as well as their activation of plateau potentials (*Lo et al., 2002*) critical for afferent refinement (*Guido, 2018*). Here, we suggest a potential third role for NMDARs: the decorrelation of neurons on a fine-grain timescale while simultaneously enhancing slow correlations (*Mizuno et al., 2021*). We suggest that NMDAR dominance at the glutamatergic synapse is one of the multiple adaptations to reduce synchronization, specifically fast-correlations, among neurons. Although slow NMDAR dynamics play a crucial role in the reduction of correlation among the TC neurons, the mechanisms of the decorrelation are not a simple low-pass filtering which is independent of the receiver (TC neuron), but rather a complex interaction between NMDAR slow currents and neuron intrinsic dynamics (*Figure 2—figure supplement 4*). While not specifically modeled here, the parasitic correlations should specifically reduce the elimination and sorting of afferent axons that show some level of correlation (for example, same-eye ganglion cells) while not affecting those with very low correlation (for example, opposite eyes). This effect has been observed in the developing thalamus, where NMDAR blockade prevents on-off segregation within the same eye, but not eye-specific lamination (*Hahm et al., 1991*).

For our modeling, we examined a model of developing LGN in which the loss of NMDARs was compensated for by homeostatic increases in AMPA receptor currents to maintain a similar level of retinal drive. Blockade or knock-out of NMDARs has been shown to increase spontaneous EPSCs (*Kesner et al., 2020*), increase AMPAR-driven circuit excitability (*Kesner et al., 2020*), and increase glutamatergic synapse density (*Rocha and Sur, 1995*; *Colonnese and Constantine-Paton, 2006*), consistent with our modeling. However, it is likely that *in vivo*, there are physiological limitations that prevent the developing nervous system from fully augmenting AMPAR currents in NR1 knock-outs to levels observed in our model. In fact, NMDAR knock-out or chronic blockade sometimes results in no change in AMPAR currents or expression (*Colonnese et al., 2003*) or their delay (*Zhu and Malinow, 2002*). Nevertheless, the insights provided here are still applicable regardless of the effects of NMDAR elimination of excitability because they demonstrate how NMDARs decorrelate early activity and provide testable predictions that can be calibrated to the levels of AMPAR homeostasis observed following particular experiments.

## Are there other options to suppress parasitic correlation?

Our model indicates that a specific, beautifully choreographed sequence of events in the developing brain evolved to keep TC neurons from precise synchronization. First, the dominance of NMDAR currents interacting with intrinsic neuron dynamics decorrelates spiking in TC neurons in rapid timescales (*Figure 2*). Then, the strength of inhibition from TRN (*Figure 4*) and from the cortex (*Figure 5*) stays low until it is "safe" to increase both synaptic strengths without allowing expression of the parasitic correlations. However, would it be possible to decorrelate the dLGN network without such a complex sequence, and under which particular conditions would this sequence be advantageous?

Around eye-opening, visual cortex activity changes from unstable, discontinuous activity with synchronized activity in the coarse-grained timescale to bistable continuous activity with sparse firing in the upstate (*Colonnese and Phillips, 2018*). Similar changes happen across multiple sensory

and association cortices around birth in humans and the start of the third postnatal week in rodents (*Colonnese and Phillips, 2018*; *Chini et al., 2022*). It is believed that the developmental switch to adult-like activities is defined by the onset of fast cortical inhibition, which drives cortical networks into an asynchronous state (*Kirmse and Zhang, 2022*; *Murata and Colonnese, 2019*; *Colonnese and Phillips, 2018*). At the moment of inhibition onset, cortical networks are switched to the balanced state with chaotic, sparse firing, a well-studied regime after the seminal work of *van Vreeswijk and Sompolinsky, 1996*. The balance can be controlled by synaptic facilitation and depression observed during development (*Jia et al., 2022*).

Surprisingly, a similar inhibitory balancing by local inhibitory neurons is not used by the developing thalamus to eliminate parasitic correlations. A key reason the thalamus does not use local inhibition for decorrelation is that such a mechanism will affect all timescales, including correlations above 100ms. The mechanisms studied in this work preserve correlations induced by retinal waves but sharply attenuate correlations below the relevant range (*Figure 2*). Another reason is the differences in the dynamics of input firing rates. While in the adult cortical network, neurons receive a constant 'barrage' of excitation from the local and distant sources, during development input is pulsatile, with prolonged periods of silence followed by active bursting during a retinal wave. Taking into account, that inhibition is relatively slow before eye-opening and delayed in the cortex by about 50–100ms (*Colonnese, 2014*), the onset of the excitatory-inhibitory balance will be much slower potentially allowing highly synchronized oscillations at the beginning of each retinal wave.

## Model innovations, limitations, and predictions

A few innovative approaches were used in this study. First, we applied PCA of the database of model parameters for single neurons to show that EMOs achieved accuracy sufficient to distinguish neurons by their model parameters (*Figure 1—figure supplement 1*). We found that PCA of model parameters could be useful for qualitative assessment of EMO quality and heterogeneity of obtained models. Second, to set the strength of retinal inputs, we used the firing rate of TC neurons observed *in vivo* and 'functionally' estimated synaptic conductance at this age by assuming that total synaptic conductance is subject to homeostatic regulation and firing rate is the target activity parameter (*Riyahi et al., 2021*). Using homeostatic regulation to set synaptic conductance is critical for modeling heterogeneous networks, where a single value for synaptic conductance would not work for all neurons. Although neurons have different mean firing rates even after homeostasis converges, these firing rates have a plausible distribution around the mean firing rate. Therefore, by applying well-developed homeostatic mechanisms to a heterogeneous network, we developed a novel approach that allows the maintenance of both the heterogeneity of the neurons and the heterogeneity of firing rates in the biologically reasonable ranges. The same homeostatic regulation allowed us to study cortical and TRN feedback without individually fitting the firing rate of each TC neuron.

To our knowledge, we present here the most detailed model of developing retino-thalamic circuitry published. Even so there are a few important limitations. The amount and organization of RGC divergence has been inferred from low-resolution anterograde tracing. If early RGCs do not connect with multiple local dLGN relay neurons to the degree we propose, it will likely reduce the synchronization we observe. Because we modeled CT and TRN feedback as simple loops, we cannot account for any dynamics intrinsic to these regions. Both TRN and cortex generate activity intrinsically *in vitro* (*Pangratz-Fuehrer et al., 2007*; *Garaschuk et al., 2000*). Visual cortex in particular produces highly synchronous events that are not dependent on the retina (*Siegel et al., 2012*). These events may be independent of thalamus and involved in homeostatic regulation of cortical synapses (*Wosniack et al., 2021*). While independently generated synchronous events could increase the functional connectivity for TRN and/or cortex by a few orders of magnitude, our modeling suggests this is not enough to drive large increases in the rapid synchronization of dLGN (*Figures 4 and 5*).

Our models specifically targeted the thalamic network and studied spike correlations in different timescales in this network. The strongest model prediction is that the dominance of NMDAR currents and developmental delays in an increase of corticothalamic and reticulothalamic synapses in early development should diminish parasitic correlation while keeping correlation in the range of retinal waves intact. This prediction can be experimentally tested in a few ways. The role of NMDAR currents can be validated by a dense extracellular recording of TC neurons in NR1 (*Grin1*)-null mice, in which NMDARs are knocked out only in the thalamus but intact in the cortex (*Arakawa et al., 2014*). We

expect higher precise, but not slow, correlation in these animals than in wild type. The effects of stronger feedback connections can be potentially tested by driving the visual cortex and/or TRN with optogenetic stimulation, creating an opportunity for parasitic correlation in the enhanced, adult-like thalamocortical loop.

## Novel principle of development

Our results unexpectedly suggest that the nervous system may have evolved to avoid specific network dynamics that appear due to the unrefined connections present during development. Many afferents refine their connectivity during development, a process that allows activity and experience to influence the final circuit connectivity (*Vonhoff and Keshishian, 2017*; *Kano and Hashimoto, 2009*; *Sanes and Lichtman, 1999*). How this exuberance affects the developing circuit dynamics and how circuits select which activity to use and which to discard has not been extensively considered. Our results suggest that just as specific circuit properties have evolved to generate and transmit activity necessary for proper circuit formation, such as retinal or cochlear waves (*Elstrott and Feller, 2010*), specific mechanisms, such as the dominance of NMDAR current (*Figure 2*) or delay the development of TRN and cortical connections (*Figures 4 and 5*) developed to suppress activity that would be deleterious to the developing system. In this case, the detrimental activity is a precise parasitic correlation arising from early convergence in the unrefined inputs, but we expect there are other generators and other suppressors. Overall, we propose a potentially general principle of neurodevelopment: developing circuit and synaptic properties are balanced to optimize the transmission of informative activity as well as to suppress the dynamics which appear due to the incompleteness of network connections and which cannot be used to properly refine the network.

## Materials and methods
### Experimental procedures

Recordings of dLGN neurons were made as part of the baseline characterization in published experiments. Animal care and procedures were in accordance with the Guide for the Care and Use of Laboratory Animals (NIH) and approved by the Institutional Animal Care and Use Committee at the University of Louisville (Protocol IACUC 21937). *In vitro* whole-cell patch recordings were obtained from mice of either sex dLGN neurons. Borosilicate glass pipettes were pulled from a vertical puller (Narishige) and had a tip resistance of 5–10 MOhm when filled with an internal solution containing the following:117 mM K-gluconate, 13.0 mM KCl, 1 mM MgCl2, 0.07 mM CaCl2, 0.1 mM EGTA, 10 mM HEPES, 2 mM Na-ATP, and 0.4 mM Na-GTP. The pH and osmolality of the internal solution were adjusted to 7.3 and 290 mOsm, respectively. Brain slices were transferred to a recording chamber that was maintained at 35°C and continuously perfused with ACSF (3.0 ml/min). Neurons were visualized using an upright microscope (BX51W1, Olympus) equipped with differential interference contrast optics. Whole-cell recordings were obtained using a Multiclamp 700B amplifier (Molecular Devices), signals were sampled at 2.5–5 kHz, low-pass filtered at 10 kHz using a Digidata 1320 digitizer and stored on a computer for subsequent analyses using pClamp software (Molecular Devices). Access resistance (15MOhm) was monitored continuously throughout the experiment, and neurons in which access resistance changed by 20% were discarded. A 10 mV junction potential was subtracted for all voltage recordings.

### Neuron optimization pipeline

Each neuron model consists of a somatodendritic single-segment compartment and a multisegmental axonal compartment. The number of segments in the axonal compartment is set to an odd value so that segments are no longer than 0.1 of the AC length constant at 100 Hz (*Hines and Carnevale, 2001*). In the somatodendritic compartment, there are nine cross-membrane channels: leak current, fast sodium and delayed rectifier potassium currents, persistent sodium current, transient and depolarization-activated potassium current, L-type calcium current, low-threshold calcium current, SK-type calcium-activated potassium current, and nonselective voltage-gated cation current. The complete list of model parameters open for adjustment by EMO is given in the *Table 1*.

Two methods of multiobjective evolutionary optimization (EMO) were used to reproduce the dynamics of TC neurons in the biophysical model. Namely: genetic algorithms with nondominated

**Table 1.** Open Model Parameters.

| Parameter(s) | Scale | minimal boundary | maximal boundary |
|---|---|---|---|
| soma.L | linear | 20. | 200. |
| soma(0.5).pas.e | linear | –55. | –90. |
| soma(0.5).pas.g | logarithmic | 1e-7 | 1e-1 |
| soma(0.5).TC_HH.gk_max | logarithmic | 1e-7 | 1e-1 |
| soma(0.5).TC_HH.gna_max | logarithmic | 1e-7 | 1e-1 |
| soma(0.5).TC_HH.vtraub | linear | –70. | 20. |
| soma(0.5).TC_HH.vtraub2 | linear | –70. | 20. |
| soma(0.5).SK_E2.gSK_E2bar | logarithmic | 1e-7 | 1e-1 |
| soma(0.5).SK_E2.zTau | linear | 1. | 500. |
| soma(0.5).TC_iT_Des98.shift | linear | –25. | 25. |
| soma(0.5).TC_iT_Des98.actshift | linear | –25. | 25. |
| soma(0.5).TC_iT_Des98.pcabar | logarithmic | 1e-7 | 5e-1 |
| soma(0.5).TC_ih_Bud97.gh_max | logarithmic | 1e-7 | 5e-1 |
| soma(0.5).TC_ih_Bud97.e_h | linear | –50. | 0. |
| soma(0.5).TC_Nap_Et2.gNap_Et2bar | logarithmic | 1e-7 | 5e-1 |
| soma(0.5).TC_cad.taur | linear | 2. | 30. |
| soma(0.5).TC_cad.gamma | logarithmic | 1e-5 | 1e-1 |
| soma(0.5).TC_iA.gk_max | logarithmic | 1e-7 | 1e-1 |
| soma(0.5).TC_iL.pcabar | logarithmic | 1e-7 | 1e-1 |
| soma.cao | linear | 1. | 6. |
| soma.ena, axon.ena | linear | 40. | 65. |
| soma.ek, axon.ek | linear | –65. | –110. |
| axon.diam | linear | from mrth import *.5 | 5. |
| axon.L | linear | 100. | 1000. |
| axon(0.5).TC_HH.gk_max | logarithmic | 1e-7 | 1. |
| axon(0.5).TC_HH.gna_max | logarithmic | 1e-7 | 1. |
| axon(0.5).TC_HH.vtraub | linear | –70. | 20. |
| axon(0.5).TC_HH.vtraub2 | linear | –70. | 20. |
| axon.Ra, soma.Ra | linear | 20. | 120 |

sorting (NSGA2) (**Deb et al., 2002**; **Deb, 2001**), implemented in the Python library *inspyred* by Dr. Aaron Garrett (**Garrett, 2012**; **Tonda, 2020**) and developed in-house genetic algorithm with Krayzman's adaptive multiobjective optimization (see Apendix - Genetic algorithm with Krayzman's adaptive multiobjective optimization (KAMOGA)). NSGA2 uses the Pareto archival strategy, selecting one model over another if it is better than or equal to the other model in all fitness functions and strictly better in at least one fitness function. This criterion is used to determine whether an individual model is selected for entry into the final archive, which only occurs if the model is at least as good as the other models in the archive. NSGA2 performs well for single and multicompartment neuron models but requires pre-fitted passive cable properties, which is an additional step in the optimization procedure (**Neymotin et al., 2017**).

In contrast, KAMOGA re-adjusts the weights of fitness functions so that the resulting distribution of overall fitness in the generation correlates with distributions of individual fitness functions (**Eremenko**

*et al., 2019*). KAMOGA avoids over- or under-representing individual fitness functions in the overall fitness and balances EMO objectives during the optimization.

EMO ran 1024 generations of 240 models each (245,760 models in total) either for NSGA2 or KAMOGA. The light elitist selection was used for KAMOGA, holding 30 best models in the next generation from the previous one (12%). Two sets of fitness functions were used: absolute difference in the number of spikes and Euclidean distance between voltages samples for a recorded neuron and a model. Both fitness functions were applied for selected traces in the current-clamp protocol, keeping the total number of fitness functions between 40 and 54, for 20–27 traces, correspondingly.

After GA finishes, all obtained models are re-evaluated and sorted again. The top 510 models (0.2 %) are selected for automatic validation. Validation checks that model parameters are biophysically correct, namely: the length of an axon is longer than a diameter, the diameter of the axon is smaller than soma diameter, the depth of $Ca^{2+}$ buffer is smaller than soma size, and that spikes propagate without decrement through the axon. Then the sets of the model parameters that pass validation were evaluated by a human. The pipeline produces from 10 s to a few hundred models for each recorded neuron.

Note that EMO searches in logarithmic space for some model parameters, while the others have linear scale (see second column in the *Table 1*). The choice of scaling depends on the ratio between minimal($p_{min}$) and maximal($p_{max}$) boundaries for a particular parameter. For example, EMO performs better in a linearly scaled space for a reversal potential($p_{max}/p_{min} \approx \mathcal{O}(1)$), while it shows better results in logarithmic scaled space for a channel conductance ($p_{max}/p_{min} \approx \mathcal{O}(10^6)$).

The code for both EMO and supporting tools are available through GitHub (copy archived at *Tikidji-Hamburyan, 2022*).

The obtained database was analyzed in two ways. First, we analyzed the resulting database by obtaining Gaussian estimator with the bandwidth defined by Scott's Rule for each parameter separately. We used the algorithm of the kernel density estimator implemented in the *scipy* Python library (*Figure 1B*). Second, we computed the first 5 principal components of all models in the database using *sklearn* Python library (*Figure 1—figure supplement 1*).

## dLGN network model

Parameter sets for individual TC neurons are randomly picked from the database. Neurons are deployed on 7x16 hexagonal lattice, 112 neurons in total. An *ex vivo* recording electrode lattice (usually square) is scaled and center to the neuron lattice. Probability of connection between GCs and TC neurons is defined as $\exp(-|\mathbf{r}_{GC}, \mathbf{r}_{TC}|^2/\sigma^2)$ while the synaptic conductance is defined as $g_0 \exp(-|\mathbf{r}_{GC}, \mathbf{r}_{TC}|^2/\sigma^2)$, where $\sigma$ is a convergence parameter, $g_0$ is the minimal synaptic conductance needed for triggering a single spike in neuron model for given parameter set and NMDAR/AMPAR conductance ratio, $|\mathbf{r}_{GC}, \mathbf{r}_{TC}|$ is a distance between GC and TC neurons.

Each synapse is modeled as a two-stage process. The first is a simplified Tsodyks and Markram model (*Tsodyks et al., 2000*) implemented by Dr. Ted Carnevale (see here). The parameter of presynaptic single spike depression ($u_0$) was set to 0.3 for match a paired-pulse ratio of 0.73 (*Chen and Regehr, 2000*). Time profiles for both NMDAR and AMPAR are modeled as double-exponential synapses with time constants 1ms rise, 2.2ms decay for AMPAR (*Chen and Regehr, 2000*), and 1ms rise, 150ms decay for NMDAR (*Chen and Regehr, 2000*; *Dilger et al., 2015*). Proportion of NMDAR and AMPAR conductance was derived from the peak-to-peak ratio of currents $i_{AMPA}/i_{NMDA} = \beta_{P7} = 0.78 \pm 0.09$ (*Shah and Crair, 2008*). The current in voltage clamp experiment is $i_{vl} = g_{vl} * (E - V_{vl})$ where $E = 0$ reversal potential for NMDAR and AMPAR, and $v_{VL}$ potential for voltage clamp. Substitute the current and voltage to $g_{vl} = i_{vl}/v_{vl}$ we can get a fraction of AMPAR to NMDAR conductance as

$$\frac{g_{AMPA}}{g_{NMDA}} = \frac{i_{AMPA}}{i_{NMDA}} \frac{\Delta_{NMDA}}{\Delta_{AMPA}},$$

where $\Delta_{NMDA}$ and $\Delta_{AMPA}$ are differences between the holding and reversal potentials for NMDAR and AMPAR currents. For both *Shah and Crair, 2008* and *Chen and Regehr, 2000* experiments $\frac{\Delta_{NMDA}}{\Delta_{AMPA}} = 40\text{mV} / 70\text{mV} = 0.57$, and ratio of NMDAR to AMPAR conductance, therefore

$$\frac{g_{NMDA}}{g_{AMPA}} = \frac{1}{0.57\beta_{P7}} \approx 2.25$$

Note that *Dilger et al., 2015* estimated $\beta_{P10} \approx 1$ which give $g_{NMDA}/g_{AMPA} = 1/0.57 \approx 1.75$, while the same value from *Chen and Regehr, 2000* $\beta_{P10} = 0.5 \pm 0.1$ gives $g_{NMDA}/g_{AMPA} = 1(0.57\ 0.5) \approx 3.5$. In this model, we used $g_{NMDA}/g_{AMPA} = 2.25$ value. We assume that 13% of NMDAR current is conducted by $Ca^{2+}$ ions and, therefore, we add this current as calcium current computed as Goldman–Hodgkin–Katz equation.

Because neurons in the dLGN model can show a wide range of firing rates, specifically just after model initiation, linear firing-rate homeostasis does not show robust results. We used nonlinear firing-rate homeostasis, which scales all synapses at the given neuron as

$$g_{i,j} = g_{i,j} \left( 1 + \tanh \alpha(r_0 - r_i) \right)_{g_{min}}^{g_{max}}$$

where $g_{i,j}$ is synaptic conductance from $j^{th}$ rGC neuron to $i^{th}$ TC neuron limited by maximal $g_{max} = 10g_0$ and minimal $g_{min} = 0$ boundaries, $\alpha = 0.05$ is a gain of homeostasis at target firing rate $r_0$, and $r_i$ is a current firing rate of $i^{th}$ TC neuron. The current firing rate $r_i$ is computed every 120 s. By the end of that interval, synaptic weights are updated. The mean firing rate is estimated over 27-minute intervals, and the simulation continues until the mean firing rate will not reach the target firing rate with 10% margins. Homeostasis requires from 11 to 22 intervals to converge to the target firing rate, setting overall simulation time from 5 to 10 hr of model time.

The TRN inhibitory feedback is sketched as a non-specific inhibitory loop with significant delay. The GABA current is modeled as double exponential synapse with 5ms rise and 50ms decay time constants and –70 mV reversal potential. Note that shunting inhibition shows a much weaker effect than hyperpolarizing inhibition, therefore, in some test simulations (not shown), we used up to –90 mV reversal potential to justify results reported here.

Cortical excitatory feedback is modeled as a non-specific excitatory loop with significant delay. NMDAR and AMPAR currents have the same parameters as in rGC projections, except presynaptic single spike depression ($u_0$), which was set to 0.7 to match much stronger paired-pulse depression in cortical axons (WG unpublished data). For the complete connectivity models, the delay for TRN connections was strictly longer than for cortical connections because there are no dLGN→TRN connections at this age, and excitation should pass the cortex before reaching TRN (*Figures 4A and 5A*).

For both TRN inhibitory feedback (*Figure 4*) and fully connected network (*Figure 5*), feedback was added as all-to-all synapses onto the same dLGN network. Synaptic conductance and delays were homogeneous in these feedback loops. However, for several models, where high correlation is observed, we ran simulations with jitter in feedback spikes. In this case, we randomly perturbed delay in the range ±20% to mimic possible desynchronized activity in cortex and TRN. This heterogeneity of feedback did not appreciably change the result (data not shown).

## Quantification of spike correlation

To compute spike correlation in all figures except *Figure 2C* and in *Figure 2—figure supplement 3*, the spike time histograms (STHs) with 1ms time bin were computed from spike trains of each neuron. Then each STH was convolved with Mexican-hat-like kernel, which is a difference between two normalized by space under the curve Gaussian curves with 20ms sigma and 80ms sigma. Standard Pearson's correlation was computed for each pair of smoothed STHs using the *corrcoef* routine from *numpy* library. A histogram of correlations was computed for each network and an average distribution over 10 models was computed for each convergence factor (*Figure 2A2 and B2*). The mean network correlation and standard deviation then plotted on *Figure 2A3 and B3*. Note that the Mexican-hat kernel allows a strong negative correlation when two neurons spiking in antiphase on the peak-frequency of the kernel, the property that a simple Gaussian kernel does not have.

To estimate correlation in different time scales (*Figure 2C*) we compute the same mean network correlation but for Mexican-hat kernels comprised a positive Gaussian with sigma (10, 14, 20, 29, 41, 59, 84, 121, 172, 246, 350, 500) ms and negative Gaussian with (40, 56, 80, 116, 164, 236, 336, 484, 688, 984, 1400, 2000) ms, respectfully for each correlation timescale. To have a one number for horizontal axis, we report (*Figure 2 C1–C3*) the size of time window where the Mexican-hat kernel is positive, that is (24, 34, 48, 70, 98, 142, 204, 292, 412, 560, 726, 886) ms windows for the positive and negative components above.

## Quantification of mutual information

To compute mutual information, we first filter burst onset time, as in the original paper **Butts and Rokhsar, 2001**, but with a shorted threshold of 0.1 s for burst onset, as TC neurons have higher firing rates than rGC neurons at P7-P10 (**Murata and Colonnese, 2018**). We then compute distributions of interburst intervals for each pair of neurons in our model, and reconstruct conditional distribution $p(\Delta t | r)$. Note that we will refer to both interburst intervals and interspike intervals as ISI because the results for this analysis are almost identical, but computations with interburst intervals can be performed on a regular computer with 64 Gb memory, while computations with interspike intervals need more than 128 Gb memory and were performed on high-memory nodes of the High-Performance Computing Cluster at the George Washington University (**Computing, 2020**). We then use the Shannon Mutual Information (MI), a quantitative measure of the interdependence of neuron separation ($r$) and ISI ($\Delta t$), using of the conditional distributions $p(\Delta t | r)$ as follows:

$$I[r, \Delta t] = \sum_r p(r) \sum_{\Delta t} p(\Delta t | r) \log_2 \left[ \frac{p(\Delta t | r)}{p(\Delta t)} \right]$$

where $p(r)$ is the prior distribution, representing the probability that two neurons chosen at random are in distance $r$ apart, and $p(\Delta t) = \sum_r p(r) p(\Delta t | r)$ overall distribution of ISIs.

## Simulation software and model availability

The models of individual TC neurons for EMO and the entire dLGN model were implemented as a Python-3 scripts for running the NEURON simulator (**Hines and Carnevale, 2001**). Simulations were sped up by employing multithreading through the NEURON's ParallelContext mechanisms. Python-3 code for the optimization pipeline and the network model is publicly available via the ModelDB website (**McDougal et al., 2017**) here.

## Acknowledgements

This work was supported by NIH grants R01EY022730 and R01NS106244 to MTC, and EY012716 to WG. This work was completed in part with resources provided by the High Performance Computing Cluster at the George Washington University, Information Technology, Research Technology Services (**Computing, 2020**) and Neuroscience Gateway Portal (**Carnevale et al., 2014**).

## Additional information

### Funding

| Funder | Grant reference number | Author |
|---|---|---|
| National Eye Institute | R01EY022730 | Matthew T Colonnese |
| National Institute of Neurological Disorders and Stroke | R01NS106244 | Matthew T Colonnese |
| National Eye Institute | EY012716 | William Guido |

The funders had no role in study design, data collection and interpretation, or the decision to submit the work for publication.

### Author contributions

Ruben A Tikidji-Hamburyan, Conceptualization, Software, Formal analysis, Validation, Investigation, Visualization, Methodology, Writing – original draft, Writing – review and editing; Gubbi Govindaiah, Data curation; William Guido, Data curation, Funding acquisition; Matthew T Colonnese, Conceptualization, Supervision, Funding acquisition, Writing – original draft, Project administration, Writing – review and editing

### Author ORCIDs

Ruben A Tikidji-Hamburyan http://orcid.org/0000-0002-0309-2129

Matthew T Colonnese [iD] http://orcid.org/0000-0002-2480-1270

### Ethics

Animal care and procedures were in accordance with The Guide for the Care and Use of Laboratory Animals (NIH) and approved by the Institutional Animal Care and Use Committee at The University of Louisville (Protocol IACUC 21937).

### Decision letter and Author response

Decision letter https://doi.org/10.7554/eLife.84333.sa1
Author response https://doi.org/10.7554/eLife.84333.sa2

---

## Additional files

### Supplementary files
• MDAR checklist

### Data availability

All models are publicly available via the ModelDB website. The database with 286 models of TC neurons at P7 is avaliable on Zenodo. The code for both EMO and supporting tools are available on GitHub (copy archived at *Tikidji-Hamburyan, 2022*).

The following datasets were generated:

| Author(s) | Year | Dataset title | Dataset URL | Database and Identifier |
|---|---|---|---|---|
| Tikidji-Hamburyan RA | 2022 | Model parameter | https://doi.org/10.5281/zenodo.7312024 | Zenodo, 10.5281/zenodo.7312024 |
| Tikidji-Hamburyan R | 2023 | Decorrelation in the developing visual thalamus | http://modeldb.yale.edu/267589 | ModelDB, 267589 |

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

## Appendix 1

### Genetic algorithm with Krayzman's adaptive multiobjective optimization (KAMOGA)

This EMO is built upon the algorithm for adjustment of weights for individual fitness (cost, objective, penalty) functions developed by *Eremenko et al., 2019* for Reverse Monte Carlo optimization in physics. For each $i^{th}$ parameter set, a vector of fitness functions $\mathbf{F}_i = \{F_{i,j}\}_j$ is computed, where $F_{i,j}$ is the $j^{th}$ fitness function. For each generation of the genetic algorithm (GA), $\mathbf{F}_i$ are combined into a fitness matrix $\mathbf{F}$, and the final single-value fitness for each parameter set ($s_i$) within the generation is obtained by multiplication of the fitness matrix by the weight vector $\mathbf{s} = \mathbf{F} \cdot \mathbf{w}$. At the beginning, weights are initiated as the normalized inverse variance for each fitness function, i.e. $w_j = 1/var_i F_{i,j}$, and then $w_j = w_j/\max \mathbf{w}$, where subscribed index (i.e., $i$) denotes the index along which variance is computed.

For each next generation, two more vectors are computed: the first is the same $\mathbf{s}$ as above, and the second is a vector of correlations between individual fitness functions within the generation and vector $\mathbf{s}$:

$$c_j = \mathrm{corr}_i F_{i,j} s_i$$

where $\mathrm{corr}_i$ denotes the Pearson correlation coefficient computed along $i^{th}$ index.

Next, iteratively, the ratio between minimal and maximal correlation is computed $r = \min \mathbf{c}/\max \mathbf{c}$, and until this ratio is below a threshold ($\theta_r$) the algorithm performs an iteration procedure. For each iteration, the weight with minimal correlation increases, while the weight with maximal correlation decreases inversely to the number of fitness functions:

$$w_{\mathrm{argmin}\ c_j} = w_{\mathrm{argmin}\ c_j}(1 + 2/n)$$
$$w_{\mathrm{argmax}\ c_j} = w_{\mathrm{argmax}\ c_j}(1 - 1/n)$$

where $n$ is the total number of fitness functions for MO. Then the weight vector is renormalized, and a new vector of single-value fitness for each parameter set ($\mathbf{s}$) and a new correlation vector comprised by correlations for each fitness function ($\mathbf{c}$) are computed as above. Iterations stop when $r > \theta_r$. The iteration procedure is not always converging, and $r$ may systematically decrease. Therefore, there is a cap on the maximal number of iterations in which $r$ decreases sequentially, and it is set to 300. If $r$ does not increase for 300 iterations in a row, all weights are reset ($w_j = 1/var_i F_{i,j}$) and renormalized ($w_j = w_j/\max \mathbf{w}$).

A critical property of KAMOGA can be seen from the point of view of the gradient descent procedure in machine learning. KAMOGA decreases weights for fitness functions that correlate and increases weights for fitness functions that anticorrelate or uncorrelate with the weighted sum. Therefore, KAMOGA is a gradient descent toward a "flatter surface" of fitness surface in n-dimensional space. It prevents the over- or under-representation of the individual fitness function in the weighted sum and equilibrates fitness contributions dynamically with the progression of GA. Interestingly, with $\theta_r \geq 0$, the algorithm guarantees that all fitness functions correlate with the weighted sum.

We used GA with elitist selection and standard adaptive mutation to avoid convergence to a local minima. However, GA with elitist selection should be modified to use Krayzman's algorithm. To avoid mixing fitness functions obtained with different weight vectbothors, the number of elites must be set to zero whenever the iteration procedure begins by condition $r < \theta_r$. As a result, high threshold values can force the algorithm to update weights with each generation GA, abolishing any advantages of elitism. Thus, the threshold $\theta_r$ should be chosen in such a way that GA will produce at least a few generations before it loses elites. Although higher values of the threshold $\theta_r$ may help to find better a balance between fitness functions, we used $\theta_r \approx 0$ as a better choice for GA with elitist selection.

