## [Editor Report]

The authors use detailed simulations to convincingly demonstrate that the temporal properties of synaptic transmission from retina to thalamus help to prevent short timescale correlations from hijacking the activity-dependent refinement of these circuits. These correlations are shown to be "parasitic" because although they can readily drive neural plasticity, they have little information about visual topography during the relevant period of refinement. This is an important point since it informs our understanding of activity-dependent development of neural circuits. The present study shows that it is not enough to simply posit that "neurons that wire together fire together," since some types of correlated firing are actually detrimental.

---

## [Decision Letter]

**Decision letter after peer review:**

Thank you for submitting your article "Synaptic and circuit mechanisms prevent detrimentally precise correlation in developing visual system" for consideration by *eLife*. Your article has been reviewed by 3 peer reviewers, including Sacha B Nelson as the Reviewing Editor and Reviewer #1, and the evaluation has been overseen by Andrew King as the Senior Editor. The following individual involved in the review of your submission have agreed to reveal their identity: Matthias H Hennig (Reviewer #3).

Essential revisions:

1) All of the reviewers felt that the manuscript needed to be revised to improve the clarity of the presentation so that it is more accessible to a non-specialized audience. Please see the individual reviews below.

2) The reviewers felt that the present manuscript failed to provide sufficient insight into how decorrelation was achieved in the simulations. There was confusion about the following interrelated issues:

a) are NMDA receptors in the model simply performing a low-pass filtering operation?

b) what is the precise origin of the high-frequency or "fast" correlations that are being removed?

c) in what sense, more precisely, are these "parasitic?" (the term, if retained, should be better defined, but perhaps it would be better to choose a more neutral term). One of the reviewers suggested that the problem with "referring to precise correlations as parasitic [is that] what is bad or good depends on the needs of the circuit."

d) Is it really correlations over different timescales or simply the magnitude of the correlation at a fixed window--to address this, the correlation over a broader range of different time lags should be computed.

e) Are other factors, such as the heterogeneity of the model neurons or the broad and imprecise connectivity also contributing? The contribution of at least some of these assumed circuit features should be addressed in additional simulations.

3) The paper should propose some experimental predictions of the model.

*Reviewer #1 (Recommendations for the authors):*

My major suggestion for improvement is to extensively rework the presentation to make it more accessible to non-specialists. Many of the key ideas are referred to throughout the paper but are only explained in the discussion. These need to be explained intuitively much earlier in the paper and then supported by the results of the simulations. In addition, there are many fine points that are difficult for all but the most specialized reader to follow.

Specific suggestions are below.

Title: In keeping with *eLife* policy, add a reference to "mammalian".

48 "Correlation below such timescales is likely to be damaging as it has the potential to induce synaptic plasticity based on non-informative activation."

This core point needs more explanation when it is introduced as this point will not be immediately obvious to most readers.

50 "the long timescales of development" is ambiguous. I think you mean the longer timescales of correlations present in the developing visual system, as opposed to the developmental timescale of days to weeks (mouse) or weeks to months (human).

51 For more foundational references on this issue, recognized much earlier than 2014, see references contained in Nelson and Sur 1992, https://doi.org/10.1016/0959-4388(92)90184-M

61-62: A key piece of the argument could be made a little more explicitly: Rapid timescale correlations are present due to visual stimulation and for these to accurately drive refinement, the refinement must not have already occurred before eye-opening. Either, the refinement must be gated, or the correlations which would otherwise drive it must have been prevented in some way.

Another point that is unclear here is why these finer timescale correlations are absent from retinal waves, i.e. do not convey information about the retinal position. I'm sure this has a reasonable explanation, but it's not going to be obvious to most readers. (added after reading further: I see part of the explanation is in Figure 5, perhaps it would be helpful to anticipate this result in the introduction-but even here the explanation is in terms of mutual information between wave and synaptic transmission as a function of synaptic transmission properties but does not explain the issue of what is or is not encoded in the waves. I haven't gone back and reviewed the Butts papers, but what most who have read them are likely to remember is that there is some information about topography in the waves, not the precise spatiotemporal point at which this information falls off – e.g. fact given in line 333 probably needs to be given earlier and more frequently and given some more intuitive motivation as to why it is true mechanistically).

If I am not understanding this correctly, that is further evidence that the precise argument here is not completely clear.

102: can you motivate why existing recordings were not sufficient?

139-151: The significance of the fact that the parameter distributions are not multimodal is not clear. It is also unclear how this leads to the conclusion "Thus we conclude that our database is a valid representation"… Is the fact that these models are nearby in parameter space just a validation of the fact that (a) the neurons are compact and (b) a reasonably orthogonal set of channels was chosen? I'm struggling to understand the point.

Figure 1 legend: most of the description in B should probably move to the methods (e.g. which algorithm was used, which Python library etc.).

184-202: because the homeostatic convergence is pretty critical to the endpoint synaptic weights you are looking at, this assumption could probably use a bit more justification and introduction. Even just the basic concept "Instead of setting the synaptic conductance into some specific value…" could be expanded a bit. Remember you are trying to communicate this to general readers rather than just to computational neuroscientists.

211-13: same point: explain for a more general audience why low correlation is unexpected and why it should be sensitive to convergence. Maybe just move this point (the unexpectedness) until after the contrast with 2B-AMPA only results.

293: to visual → to visualize.

Figure 5. This could use some intuitive motivation.

424-426 The definition of parasitic correlations comes very late. Consider using a more neutral term and defining it earlier to help motivate the study.

451-453 Similarly, the explanation of why waves lack short time scale correlation information about topography would benefit most readers if presented earlier.

505 "provide testable predictions" It would be helpful to say here or earlier what these predictions are and explicitly label them as such.

511-563. I do not find this section very helpful. It reads like a historical account of the process of performing the study, not like a discussion of the results.

General: the term "state-of-the-art" is overused.

*Reviewer #2 (Recommendations for the authors):*

There are my main "big picture" concerns:

1. The notion of correlation timescales (or precision) vs amplitude (or level).

The authors describe their approach for computing correlations on line 202, based on a previous study from the same lab: but the Mexican-hat-like kernel they use (difference of a 20 ms Gaussian and an 80 ms Gaussian) is already setting the timescale of the correlations, hence the only aspect that they compute and vary is the level of the correlation and not the timescale.

To provide any kind of quantification of the timescale of the correlations, the authors need to: a. Consider different timescales of the kernel, including very narrow ones, b. Consider also the correlations as a function of time lag. Note: people use a different word for this, what I mean is the cross-correlation, normalized to -1 and 1 by subtracting the mean and dividing by the standard deviation, where the typical Pearson correlation coefficient would just be the value at zero time lag.

It seems as if the authors are only measuring the correlation amplitude (level) at zero lag, which conveys information only about a single timescale (the one in their fixed kernel), and hence they cannot make claims about the connectivity convergence factor (or anything else) changing correlation timescales if they don't actually look at those correlation timescales. Basically, every measurement of correlation, e.g. Figure 2, 3, 5, etc. currently is about amplitudes at a fixed timescale, not about timescales. I would suggest that the analysis is actually repeated for correlations over different timescales.

(More minor but related comment is: It is also unclear why they need to have the negative part of the kernel, i.e. why a Mexican hat and not just a single Gaussian.)

If they just want to compare their results to those of Colonnese et al. 2017, where the same method of computing correlations is used, that's fine! But then the claims in the paper should be changed, it should be about amplitude or levels of correlations at the fixed timescales being determined by the kernel matching the data, and not about timescales of correlations.

2. The need to use a biophysical model of the thalamus.

The claim that broad and imprecise connectivity (which refers to spatial connectivity) causes correlations over millisecond timescales because of locally homogeneous synaptic currents is unwarranted. Locally homogeneous synaptic currents do not follow from the broad and precise connectivity. They follow if one assumes uniform neuronal properties. The current paper solves this by fitting biophysical thalamic neurons to data and hence generates neuronal diversity. This is an interesting result on its own, but it's an entirely different reason for the differences in synaptic currents than the broadness and precision of connectivity. What would happen if simpler neurons were used but with diverse currents?

If the main result is the presence of NMDA currents and lack of recurrent connectivity, could the same hold if simpler (single compartment, leaky IaF, or even exponential IaF) models were used for the thalamic cells? I see the value in building these databases using real data, but just to ask about synchronization and timescales of activity and plasticity, maybe a single-compartment model is enough?

Related to this: Are precise correlations generated because of the heterogeneous neurons, or because of the levels of convergence? If the claim is the latter, then is it still there for homogeneous neurons? How important is the heterogeneity among the neurons?

Along those lines: I would tone down the statement in line 92. To really claim that the authors should indeed provide evidence that not including all the detail fails to model spike correlations in the developing thalamus. Synchronization can be avoided not just by heterogeneous neurons, but also due to randomness in input drive and balanced excitation-inhibition as is the case of the classical balanced E/I networks (e.g. classical papers by van Vreeswijk and Sompolinsky and many others).

3. This may appear like a minor issue but it's actually pretty important. The authors should reconsider using the word "parasitic" when they refer to detrimental i.e. bad correlations. Parasitic means feeding someone else, but this is not the case here.

4. For the models in Figures 3 and 4, the setup is unclear. How many new neurons were added? How many TRN neurons, how many cortical neurons (or I guess synapses, not neurons)? Was there heterogeneity here?

5. What about cortically generated activity, e.g. H events as described by Siegel and Lohmann 2012? Especially for the model in the last section where external spike trains beyond the LGN input are not modeled.

6. In addition to the Butts papers, they should also discuss and cite: Gjorgjieva et al. PLoS CB 2009 which actually used real retinal waves and compared the STDP vs BTDP rule proposed by Butts 2007 to demonstrate that the timescales of development and plasticity need to be matched (both slow).

Other papers that they should also consider citing:

- Wosniack et al. 2021 *eLife* presents a similar feedforward model between the thalamus and cortex and proposes a way to decorrelate activity as a function of development.

- An alternative way to decorrelate activity via emerging inhibition is proposed by Chini et al. 2022 *eLife* and Rahmati et al. Sci Reports 2017.

- Jia et al. Communications Biology 2022 for short-term plasticity in development which could also influence correlations as it changes E/I balance (especially as the authors assume STP to operate at their synapses)

*Reviewer #3 (Recommendations for the authors):*

My main suggestion is to try to address the question of how NMDA receptors actually prevent parasitic correlations. As mentioned in the public review, perhaps the slow time constant is responsible. If this is the case, then a graded reduction of NMDA conductances (and homeostatic compensation to maintain the average firing rates) should increase their strength. So perhaps the single-cell NMDA/AMPA ratio is related to the strength of parasitic correlations? This could be an experimentally testable prediction of this work. More generally, I would find it useful if testable experimental predictions could be as specific as possible, e.g. effects of blocking (knocking out) NMDARs (subunits).

---

## [Author Response]

Reviewer #1 (Recommendations for the authors):My major suggestion for improvement is to extensively rework the presentation to make it more accessible to non-specialists. Many of the key ideas are referred to throughout the paper but are only explained in the discussion. These need to be explained intuitively much earlier in the paper and then supported by the results of the simulations. In addition, there are many fine points that are difficult for all but the most specialized reader to follow.

Thank you for this suggestion. We significantly revised the main text and added clarification and explanations for a general audience.

Specific suggestions are below.Title: In keeping with eLife policy, add a reference to "mammalian".

Thank you. We updated the title.

48 "Correlation below such timescales is likely to be damaging as it has the potential to induce synaptic plasticity based on non-informative activation."This core point needs more explanation when it is introduced as this point will not be immediately obvious to most readers.50 "the long timescales of development" is ambiguous. I think you mean the longer timescales of correlations present in the developing visual system, as opposed to the developmental timescale of days to weeks (mouse) or weeks to months (human).

Thank you, we completely rewrote the introduction. The second part of this paragraph is gone in the current version. The sentence you referred to in the first suggestion is now half of the paragraph, which reads:

“While the mechanisms of correlated activity generation are well studied, the central mechanisms by which this activity is processed and transformed into synaptic plasticity and, ultimately, circuit structure are poorly understood. One crucial question is how the correlation timescale influences developmental plasticity and how correlation timescales are regulated within early circuits because timescale can determine the plasticity mechanisms potentially available to the developing neurons (Drew and Abbott, 2006; Zenke and Gerstner, 2017). In adults, thalamic and cortical neurons produce precise correlations on the timescale of milliseconds, which depend on the visual stimulus (Butts et al., 2007). By contrast, experimental and theoretical work has shown that during development, information on visual topography during retinal waves is conveyed in coarse-grained correlation and vanishes in time windows below 100 ms (Butts and Rokhsar, 2001; Butts, 2007). Early synapses make several adaptations to accommodate the long timescales of correlation in developmental activity. These include synaptic currents, which are slower to increase the integration window (Hauser et al., 2014; Nelson and Sur, 1992), and burst-time-dependent plasticity with a characteristic time-window of around 1 second to refine topographic relationships (Butts, 2007). Although there is no good theory on why retinal ganglion cells do not correlate with better time recession to our knowledge, it is well established that the timescale of correlation should match the timescale of synaptic plasticity to robustly segregate ON and OFF pathways in later stages of retinal ways (Gjorgjieva, 2009). However, if correlation below timescales of correlated activity in the retinal ways appears in descending pathways, it is likely to mask informative activity and damage developing networks as it has the potential to induce synaptic plasticity based on non-informative activation. Because the specialized response properties such as direction selectivity and ON/OFF identity have not emerged yet (Wong, 1999), and the critical topographic information represented in the retinal waves are conveyed only above 100ms (Butts et al., 1999), any plasticity driven by rapid early correlations is likely to not result in the desired adult network connectivity.”

51 For more foundational references on this issue, recognized much earlier than 2014, see references contained in Nelson and Sur 1992, https://doi.org/10.1016/0959-4388(92)90184-M

Thank you, we have incorporated these earlier citations.

61-62: A key piece of the argument could be made a little more explicitly: Rapid timescale correlations are present due to visual stimulation and for these to accurately drive refinement, the refinement must not have already occurred before eye-opening. Either, the refinement must be gated, or the correlations which would otherwise drive it must have been prevented in some way.

Thank you for the suggestions. We have revised the introduction and results to better guide the reader and anticipate our arguments, particularly referencing Figure 3 (previously Figure 5).

“Rapid timescale correlations are absent in the cortex and thalamus during early development, emerging only after eye-opening (Colonnese et al., 2017). Millisecond, precise correlations are optimal for visual processing in adults (Butts et al., 2007), potentially rely on precise refined connections and fast synaptic transmission (see Figure 3 for an insight) and, therefore, rapid timescale correlations should be relevant only when the networks are refined and ready to operate in such timescales.”

Another point that is unclear here is why these finer timescale correlations are absent from retinal waves, i.e. do not convey information about the retinal position. I'm sure this has a reasonable explanation, but it's not going to be obvious to most readers.

That is an excellent question, which to our knowledge, has been overlooked. We know of no research or theory on why (beyond simple mechanism) the timescale of correlations in retinal waves is so slow. We have added the following sentences to the discussion:

“The reason for this slow propogation is that the wavefront is generated by the volume release of acetylecholine and its bulk diffusion (Ford et al., 2012). Why nearby rGCs are poorly correlated during waves is poorly understood. While they maintain a high firing rate (~10Hz) during a wave, this firing appears random and not correlated on a fast timescale to nearby rGCs, likely because intraretinal circuits mediating ganglion cell interaction (lateral inhibition, etc) is not yet formed.”

102: can you motivate why existing recordings were not sufficient?

The recordings were performed as routine TC neuron identification and characterization in previously published works. So they are existing recordings. We added clarification and citations in the main text.

“The neuron membrane dynamics were derived from 10 thalamocortical (TC) neurons recorded at P7-10 in vitro, recorded in standard current-clamp protocol with step current stimulation(see section for more details). The recordings were performed as routine TC neuron identification and characterization in previously published works: (Campbell et al., 2020) and (Govindaiah et al., 2020).”

139-151: The significance of the fact that the parameter distributions are not multimodal is not clear. It is also unclear how this leads to the conclusion "Thus we conclude that our database is a valid representation"… Is the fact that these models are nearby in parameter space just a validation of the fact that (a) the neurons are compact and (b) a reasonably orthogonal set of channels was chosen? I'm struggling to understand the point.

We apologize for the confusion. Based on Marder and colleagues work that a single-compartment conductance-based models can generate similar dynamics for two or more different parameter sets, we were worried that our model fits would generate multimodal distributions of model parameters with peaks at regions where model dynamics match the target dynamics of the recorded neuron. Multimodal distributions of parameter sets would indicate there are multiple very different possible combinations of parameters to model our recorded neurons. This would undermine our confidence that the modeled neurons represented real *in vivo* neurons. Thus the fact that the distribution is monomodal and that all models occupy a relatively compact region of the PCA space, suggests a unique region of parameters that match TC neurons at P7-P10, and increases confidence that our models are somewhat close to the real data. We now explain this in the first result section.

“Somatic voltage dynamics in the current-clamp protocol may be insufficient to specify the unique contributions of different ion channels to the dynamics of a conductance-based model. As shown previously by Marder and colleagues, a single-compartment conductance-based model can have similar dynamics for two sets of parameters with extreme differences in the contribution of different channels (Marder, 2011; Prinz et al., 2004, 2003). Because similar model dynamics can result from different distinct configurations of channel conductance(s), it is possible that EMO, extensively searching for model configurations, might produce acceptable models in distinctly different regions of the parameter space. If this occurred we expect to find multimodal distributions of model parameters with peaks at regions where model dynamics match the target dynamics of the recorded neuron. In this situation choosing which cluster of models best represents real neurons is difficult and would undermine confidence in our neuron models….[discussion of results]….Thus, unless a number of the parameters *in vivo* lie outside the parameter ranges considered, we conclude that our database is a consistent and likely valid representation of the real dLGN TC neurons at the specific point of their development.”

Figure 1 legend: most of the description in B should probably move to the methods (e.g. which algorithm was used, which Python library etc.).

Thank you. We moved these descriptions to the methods section.

184-202: because the homeostatic convergence is pretty critical to the endpoint synaptic weights you are looking at, this assumption could probably use a bit more justification and introduction. Even just the basic concept "Instead of setting the synaptic conductance into some specific value…" could be expanded a bit. Remember you are trying to communicate this to general readers rather than just to computational neuroscientists.

We expanded these explanations. This paragraph now reads:

“Because our model of the dLGN network consists of a heterogeneous population of neurons, randomly chosen from the database, there is no single value for synaptic conductance which could satisfy all neurons in the model. The same value for synaptic conductance can be subthreshold for some neurons in our database but drive others into a depolarization block. Therefore it is preferable to use a functional criterion to set synaptic conductance. We used the mean firing rate recorded in vivo at this age as our functional criterion.”

211-13: same point: explain for a more general audience why low correlation is unexpected and why it should be sensitive to convergence. Maybe just move this point (the unexpectedness) until after the contrast with 2B-AMPA only results.

We expanded explanations and added references to the homogeneous population.

“While biologically accurate, the level of correlation is unexpectedly low given the levels of input convergence and showed a surprising insensitivity to the convergence parameter σ. With increasing convergence, neurons receive more synaptic inputs from the same rGCs. Therefore, the spike sources of the synaptic currents overlap, and the overall activation of the population becomes more from 1-3 to 20 and should dramatically increaseσ correlation within the population. homogeneous. For the range of convergences = [1,9], the number of inputs varies Even more surprising, such a low correlation does not result from population heterogeneity. Homogeneous networks populated with the same randomly chosen TC neuron model from the database show a similar low correlation and insensitivity to the convergence (Figure 2—figure supplement 1, left). … Finally, homogeneity strongly affects models with only fast AMPA receptors (Figure 2—figure supplement 1, right). Although correlation in homogeneous networks is approximately the same for adult convergence, variability quickly increases as convergence approaches the number of connections in early development and can be more than twofold for retinal σ = 9. The dominance of fast AMPA receptors in the early stages of development (σ = [4, 9]) should induce fast correlations (which result from overlaps in unrefined. projections) but does not impose additional correlation for the adult convergence.

293: to visual → to visualize.

Thank you, we corrected it.

Figure 5. This could use some intuitive motivation.

We moved Figure 5 just after Figure 2, as the second reviewer suggested (now it is Figure 3). We agree that Figure 3 clearly shows what happens with spatial information and why parasitic correlation "drains" the ability of networks to convey spatial information by adding irrelevant correlation.

424-426 The definition of parasitic correlations comes very late. Consider using a more neutral term and defining it earlier to help motivate the study.

Thank you. In the introduction, we now define parasitic correlations in the introduction, and justify its usage in the discussion.

Intro:

“In total our results suggest a novel developmental principle: the properties of early synapses and circuits are tuned to eliminate potentially detrimental correlations that are the result of refining networks because they could damage network formation. We term such correlations ``parasitic'' as they actively leech positional information from the developing network.”

Discussion:

“We propose that such correlations, arising as a consequence of the poorly refined immature network, can be considered ``parasitic'', analogous to ``parasitic capacitance'' (or other parasitic circuit elements) in electronics that drain ``vitals'' important to circuit function and ``parasitic vortices'' in aviation which reduce elevation force and create an additional drug, unless eliminated by curving the end of the airplane wings.”

451-453 Similarly, the explanation of why waves lack short time scale correlation information about topography would benefit most readers if presented earlier.

We are not aware of published theories of why the waves lack fast correlations. Given this ambiguity, our speculations disrupt the flow the introduction. Thus we now introduce the concept that waves are slow in contrast to fast visual correlations, which is the key point, and provide a more detailed if speculative account of the reason that retinal waves are slow in the discussion.

Intro:

In adults, thalamic and cortical neurons produce precise correlations on the timescale of milliseconds, optimal for visual processing in adults (Butts et al., 2007; Usrey and Reid, 1999), potentially rely on precise refined connections and fast synaptic transmission (see Figure 3 for an insight) and, therefore, rapid timescale correlations should be relevant only when the networks are refined and ready to operate in such timescales. Confirming this, experimental and theoretical work has shown that during development, topographic information in retinal waves is conveyed in coarse-grained correlation, maximal around 500ms and absent in time windows below 100 ms (Butts and Rokhsar, 2001; Butts, 2007), because the firing of ganglion cells within a wave is not precisely correlated (Maccione et al., 2014; Stafford et al., 2009).

Discussion:

The reason for this slow propagation is that the wavefront is generated by the volume release of acetylcholine and its bulk diffusion (Ford et al., 2012). Why nearby rGCs are poorly correlated during waves is poorly understood. While they maintain a high firing rate ( 10Hz) during a wave, this firing appears random and not correlated on a fast timescale to nearby rGCs, likely because intraretinal circuits mediating ganglion cell interaction (lateral inhibition etc) is not yet formed (Tian 2004).

505 "provide testable predictions" It would be helpful to say here or earlier what these predictions are and explicitly label them as such.

We added a subsection to the discussion and listed predictions of the model:

“Our models specifically targeted the thalamic network and studied spike correlations in different timescales in this network. The strongest model prediction is that the dominance of NMDAR currents and delays in an increase of corticothalamic and reticulothalamic synapses in early development should diminish parasitic correlation while keeping correlation in the range of retinal waves intact. This prediction can be experimentally tested in a few ways. The role of NMDAR currents can be validated by a dense extracellular recording of TC neurons in NR1 (Grin1)-null mice, in which NMDARs are knocked out only in the thalamus but intact in the cortex(Arakawa et al., 2014). We expect higher precise correlation in these animals when in wild type, but not in timescales of retinal waves. The effects of stronger feedback connections can be potentially tested by driving the visual cortex and/or TRN with optogenetic stimulation, creating an opportunity for parasitic correlation in the enhanced, adult-like thalamocortical loop.”

511-563. I do not find this section very helpful. It reads like a historical account of the process of performing the study, not like a discussion of the results.

We rewrote this section indicating clearly the innovations, limitations, and predictions of our models.

General: the term "state-of-the-art" is overused.

Thank you for this comment. We removed this expression.

Reviewer #2 (Recommendations for the authors):There are my main "big picture" concerns:1. The notion of correlation timescales (or precision) vs amplitude (or level).The authors describe their approach for computing correlations on line 202, based on a previous study from the same lab: but the Mexican-hat-like kernel they use (difference of a 20 ms Gaussian and an 80 ms Gaussian) is already setting the timescale of the correlations, hence the only aspect that they compute and vary is the level of the correlation and not the timescale.To provide any kind of quantification of the timescale of the correlations, the authors need to: a. Consider different timescales of the kernel, including very narrow ones,

Thank you for this suggestion, which led to increased evidence for a specific role of NMDARs in matching the correlation timescale to retinal information. We now provide new analysis in Figure 2 examining the effect of timescales above and below those initially examined. This data shows that NMDAR-dominance affects only precise correlations and falls off in tight correlation to retinal wave time scales.

b. Consider also the correlations as a function of time lag. Note: people use a different word for this, what I mean is the cross-correlation, normalized to -1 and 1 by subtracting the mean and dividing by the standard deviation, where the typical Pearson correlation coefficient would just be the value at zero time lag.

We respectfully disagree with the reviewer. While in recurrent networks, non-zero lag correlation is a useful metric that allows catching "propagation" of activation in the network due to feedback connections, in the feedforward network, correlation in non-zero time lag cannot reflect anything beyond neuron dynamics. Also, in feedforward networks, the zero-lag correlation could be considered as a proxy of the probability that two synapses from two neurons will be potentiated or depressed if they are paired with a postsynaptic spike. Non-zero lag correlation loses this property and might not be very informative from a circuit formation point of view. We performed the analysis and, as expected, there is no time-lag correlation in network activity beyond a few milliseconds.

**Author response image 1. sa2fig1:** 

It seems as if the authors are only measuring the correlation amplitude (level) at zero lag, which conveys information only about a single timescale (the one in their fixed kernel), and hence they cannot make claims about the connectivity convergence factor (or anything else) changing correlation timescales if they don't actually look at those correlation timescales. Basically, every measurement of correlation, e.g. Figure 2, 3, 5, etc. currently is about amplitudes at a fixed timescale, not about timescales. I would suggest that the analysis is actually repeated for correlations over different timescales.

We have updated Figure 2 to show how both convergence and NMDAR current-dominance effect correlations in different timescales.

(More minor but related comment is: It is also unclear why they need to have the negative part of the kernel, i.e. why a Mexican hat and not just a single Gaussian.)

The idea behind this is that: if two neurons are firing in anti-synchrony at a peak frequency of Mexican-hat, the spike correlation will be strongly negative (around -0.6), while for just Gaussian kernel spike correlation will be only -0.07. We added a note to this effect in the Methods section.

If they just want to compare their results to those of Colonnese et al. 2017, where the same method of computing correlations is used, that's fine! But then the claims in the paper should be changed, it should be about amplitude or levels of correlations at the fixed timescales being determined by the kernel matching the data, and not about timescales of correlations.

Overall we thank the reviewer for pointing out these oversights in our analysis and hope they find that with our addition of analysis examining the timescales of correlation, we have sufficiently supported our assertions regarding the timescales of correlation.

2. The need to use a biophysical model of the thalamus.The claim that broad and imprecise connectivity (which refers to spatial connectivity) causes correlations over millisecond timescales because of locally homogeneous synaptic currents is unwarranted. Locally homogeneous synaptic currents do not follow from the broad and precise connectivity. They follow if one assumes uniform neuronal properties.

We do not completely understand this comment. We think that the reviewer is saying that correlations increase with divergence (i.e. broad connectivity) but that the degree of this synchronization would be influenced by the heterogeneity of neuronal properties that determine the heterogeneity of the synaptic and spiking responses to this input? If this is the point we agree and it is an empirical question the degree to which each influences our model and results. In the initial submission we show a direct relationship between broad (divergent) connectivity and fast synchronization. In the revision (Supplementary Figure 2) we take the reviewer’s suggestion to examine the role of heterogeneity, which has a much smaller effect (at least for the parameters we examined) than convergence/divergence is driving precise correlations.

The current paper solves this by fitting biophysical thalamic neurons to data and hence generates neuronal diversity. This is an interesting result on its own, but it's an entirely different reason for the differences in synaptic currents than the broadness and precision of connectivity. What would happen if simpler neurons were used but with diverse currents?If the main result is the presence of NMDA currents and lack of recurrent connectivity, could the same hold if simpler (single compartment, leaky IaF, or even exponential IaF) models were used for the thalamic cells?

We did not test our results on simpler models or check at which level of model complexity the mechanism breaks. We do not think pure IaF or exponential IaF can show decorrelation in the NMDAR-dominant network. We think that the basis of the NMDAR effect is similar to those proposed by Pinsky and Rinzel (1994), that the interaction of NMDAR current and slow intrinsic dynamics makes neuron firing stochastic which would not be captured in these models. In the revision we have examined speeding and slowing of neuron dynamics (Supplementary Figure 2) to show that both increase fast-synchronization, and effect that would not be observed if the desynchronization were a simple factor of NMDAR current time course and observable with IaF models.

I see the value in building these databases using real data, but just to ask about synchronization and timescales of activity and plasticity, maybe a single-compartment model is enough?

We tried to model relay neuron membrane dynamics as simply as possible. We write

“All attempts to fit a model with a single somatodendritic compartment did not produce acceptable models, probably because axons and axon-hillocks are thicker and more electrically bound with the soma at this age.”

From biological literature, it is known that axon and soma are more electrically connected during development than in adults, due to a larger diameter of axon hillock and axon at these ages. Therefore, an axon compartment in a pen-and-ball configuration was added. We achieved a much better fit as shown in Figure 1. That indicates, in our opinion, that the pen-and-ball model is the minimal model that can relatively accurately capture the dynamics of TC neurons at P7-P10.

Related to the second part of the question, as mentioned above Supplementary Figure 2 shows that the specifics of the intrinsic neuron dynamics contribute to decorrelation.

Therefore, we believe, accurate representation of TC neuron dynamics at P7-P10 is a critical part of the result.

Related to this: Are precise correlations generated because of the heterogeneous neurons, or because of the levels of convergence? If the claim is the latter, then is it still there for homogeneous neurons? How important is the heterogeneity among the neurons?

We have analyzed this in the revision (Supplementary Figure 2). We find that precise correlations exist among homogeneous neurons and are sensitive to convergence showing that it is convergence not heterogeneity that determines correlation. As expected, the amount of correlation in homogeneous models depends on the dynamics of the particular neuron selected to populate that model. Interestingly we find that homogeneous and heterogeneous populations are similar when NMDAR+AMPAR currents are used. Therefore, heterogeneity in the model is important to accurately reflect correlation levels but the decorrelative effect of NDMARs is not dependent on heterogeneity per se.

Along those lines: I would tone down the statement in line 92. To really claim that the authors should indeed provide evidence that not including all the detail fails to model spike correlations in the developing thalamus.

We have revised this statement to focus on the advantages of our approach, rather than arguing it is the only way.

Synchronization can be avoided not just by heterogeneous neurons, but also due to randomness in input drive and balanced excitation-inhibition as is the case of the classical balanced E/I networks (e.g. classical papers by van Vreeswijk and Sompolinsky and many others).

This is an interesting point, raising the notion of why does the developing nervous system not use blanched E/I to desynchronize the network? To answer this we have added a section in the discussion "Are there other options to suppress parasitic correlation?". We argue:

“A key reason the thalamus does not use local inhibition for decorrelation is that such a mechanism will affect all timescales, including correlations above 100ms. The mechanisms studied in this work preserve correlations induced by retinal waves but sharply attenuate correlations below the relevant range (Figure 2). Another reason is the differences in the dynamics of input firing rates. While in the adult cortical network, neurons receive a constant “barrage” of excitation from the local and distant sources, during development input is pulsatile, with prolonged periods of silence followed by active bursting during a retinal wave. Taking into account, that inhibition is relatively slow before eye-opening and delayed in the cortex by about 50-100ms (Colonnese, 2014), the onset of the excitatory-inhibitory balance will be much slower allowing potentially highly synchronized oscillations at the beginning of each retinal wave.”

3. This may appear like a minor issue but it's actually pretty important. The authors should reconsider using the word "parasitic" when they refer to detrimental i.e. bad correlations. Parasitic means feeding someone else, but this is not the case here.

We thank the reviewer for reminding us that parasitic has multiple meanings in different fields. We have drawn inspiration from electronics and engineering terminology here. We now provide great justification of this choice in the discussion:

“We propose that such correlations, arising as a consequence of the poorly refined immature network, can be considered “parasitic”, analogous to “parasitic capacitance” (or other parasitic circuit elements) in electronics that drain “vitals” important to circuit function and “parasitic vortices” in aviation which reduce elevation force and create an additional drug, unless eliminated by curving the end of the airplane wings.”

4. For the models in Figures 3 and 4, the setup is unclear. How many new neurons were added? How many TRN neurons, how many cortical neurons (or I guess synapses, not neurons)? Was there heterogeneity here?

Thank you for catching the incompleteness of the methods section. We added a paragraph explaining the model set up here.

“For both TRN inhibitory feedback (Figure 4) and fully connected network (Figure 5), feedback was added as all-to-all synapses onto the same dLGN network. Synaptic conductance and delays were homogeneous in these feedback loops. However, for several models, where high correlation is observed, we ran simulations with jitter in the feedback spikes. In this case, we randomly perturbed delay in the range +/20% to mimic possible desynchronized activity in cortex and TRN. This heterogeneity of feedback did not appreciably change the result (data not shown).”

5. What about cortically generated activity, e.g. H events as described by Siegel and Lohmann 2012? Especially for the model in the last section where external spike trains beyond the LGN input are not modeled.

Thank you for this suggestion to address the important point of activity generated independently of retina and thalamus. Since we have removed the last model we think a detailed discussion of H events is not necessary, but we have added discussion of this to the limitations to the model:

“Because we modeled CT and TRN feedback as simple loops, we cannot account for any dynamics intrinsic to these regions. Both TRN and cortex generate activity intrinsically in vitro (Pangratz-Fuehrer et al., 2007; Garaschuk et al., 2000). Visual cortex *in vivo* produces highly synchronous events that are not dependent on the retina (Siegel et al., 2012). These events may be independent of thalamus and involved in homeostatic regulation of cortical synapses (Wosniack et al., 2021). While independently generated synchronous events could increase the functional connectivity for TRN and/or cortex by a few orders of magnitude, our modeling suggests this is not enough to drive large increases in the rapid synchronization of dLGN (Figures 4, 5).”

6. In addition to the Butts papers, they should also discuss and cite: Gjorgjieva et al. PLoS CB 2009 which actually used real retinal waves and compared the STDP vs BTDP rule proposed by Butts 2007 to demonstrate that the timescales of development and plasticity need to be matched (both slow).Other papers that they should also consider citing:- Wosniack et al. 2021 eLife presents a similar feedforward model between the thalamus and cortex and proposes a way to decorrelate activity as a function of development.- An alternative way to decorrelate activity via emerging inhibition is proposed by Chini et al. 2022 eLife and Rahmati et al. Sci Reports 2017.- Jia et al. Communications Biology 2022 for short-term plasticity in development which could also influence correlations as it changes E/I balance (especially as the authors assume STP to operate at their synapses)

Thank you for the suggestion. We added all citations and discussed them except Rahmati et al. Sci Reports 2017, summarized and extended in the review Kirmse and Zhang Cell Reports 2022 cited previously.

Reviewer #3 (Recommendations for the authors):My main suggestion is to try to address the question of how NMDA receptors actually prevent parasitic correlations. As mentioned in the public review, perhaps the slow time constant is responsible. If this is the case, then a graded reduction of NMDA conductances (and homeostatic compensation to maintain the average firing rates) should increase their strength. So perhaps the single-cell NMDA/AMPA ratio is related to the strength of parasitic correlations? This could be an experimentally testable prediction of this work. More generally, I would find it useful if testable experimental predictions could be as specific as possible, e.g. effects of blocking (knocking out) NMDARs (subunits).

Thank you for these recommendations. In the revision we use two approaches to investigate the role of NMDARs in suppression of parasitic correlation. We investigated the role of the postsynaptic cell dynamics in the effect to determine if it is a simple property of the slow NMDAR current. The results are presented in a Supplement to Figure 2. Interestingly we find that both slowing down and speeding the membrane dynamics increases fast-correlation. This means that an interaction of developmentally-specific neuron dynamics with NMDAR dynamics, not a simple receptor autonomous property, drives decorrelation. We further investigated the NMDAR/AMPAR parametrically, and the results are included in the Supplement to Figure 2. As predicted the strength of fast correlations is inversely correlated with this ratio. We have added a section for first order tests of our modeling in the Discussion section:

“Our models specifically targeted the thalamic network and studied spike correlations in different timescales in this network. The strongest model prediction is that the dominance of NMDAR currents and delays in an increase of corticothalamic and reticulothalamic synapses in early development should diminish parasitic correlation while keeping correlation in the range of retinal waves intact. This prediction can be experimentally tested in a few ways. The role of NMDAR currents can be validated by a dense extracellular recording of TC neurons in NR1 (Grin1)-null mice, in which NMDARs are knocked out only in the thalamus but intact in the cortex(Arakawa et al., 2014). We expect higher precise, but not slow, correlation in these animals than in wild type. The effects of stronger feedback connections can be potentially tested by driving the visual cortex and/or TRN with optogenetic stimulation, creating an opportunity for parasitic correlation in the enhanced, adult-like thalamocortical loop.”